# Pathogenic Huntingtin aggregates alter actin organization and cellular stiffness resulting in stalled clathrin-mediated endocytosis

Surya Bansi Singh[1,2], Shatruhan Singh Rajput[3,4], Aditya Sharma[5], Sujal Kataria[3], Priyanka Dutta[1], Vaishnavi Ananthanarayanan[6], Amitabha Nandi[7], Shivprasad Patil[3], Amitabha Majumdar[1]*, Deepa Subramanyam[1]*

[1]National Centre for Cell Science, SP Pune University Campus, Pune, India; [2]SP Pune University, Pune, India; [3]Indian Institute of Science Education and Research, Pune, India; [4]Department of Biochemistry, University of Cambridge, 80 Tennis Court Road, Cambridge, United Kingdom; [5]Department of Computer Science and Engineering, Indian Institute of Technology Bombay, Powai, Mumbai, India; [6]EMBL Australia Node in Single Molecule Science, School of Biomedical Sciences, University of New South Wales, Sydney, Australia; [7]Department of Physics, Indian Institute of Technology, Bombay Powai, Mumbai, India

*For correspondence:
amitavamajumdar@gmail.com (AM);
deepa@nccs.res.in; deepa. subramanyam@gmail.com (DS)

Competing interest: The authors declare that no competing interests exist.

**Abstract** Aggregation of mutant forms of Huntingtin is the underlying feature of neurodegeneration observed in Huntington's disorder. In addition to neurons, cellular processes in non-neuronal cell types are also shown to be affected. Cells expressing neurodegeneration–associated mutant proteins show altered uptake of ligands, suggestive of impaired endocytosis, in a manner as yet unknown. Using live cell imaging, we show that clathrin-mediated endocytosis (CME) is affected in *Drosophila* hemocytes and mammalian cells containing Huntingtin aggregates. This is also accompanied by alterations in the organization of the actin cytoskeleton resulting in increased cellular stiffness. Further, we find that Huntingtin aggregates sequester actin and actin-modifying proteins. Overexpression of Hip1 or Arp3 (actin-interacting proteins) could restore CME and cellular stiffness in cells containing Huntingtin aggregates. Neurodegeneration driven by pathogenic Huntingtin was also rescued upon overexpression of either Hip1 or Arp3 in *Drosophila*. Examination of other pathogenic aggregates revealed that TDP-43 also displayed defective CME, altered actin organization and increased stiffness, similar to pathogenic Huntingtin. Together, our results point to an intimate connection between dysfunctional CME, actin misorganization and increased cellular stiffness caused by alteration in the local intracellular environment by pathogenic aggregates.

## Editor's evaluation

Singh et al. present an important manuscript showing that the aggregation of mutant Huntingtin in Huntington's disorder affects clathrin-mediated endocytosis, actin organization, and cellular stiffness in non-neuronal cells, which can be partially restored by overexpressing actin-interacting proteins like Hip1 or Arp3. Hip1 or Arp3 also rescue neurodegeneration driven by mutant forms of Huntingtin. The study provides interesting insights into the interplay between Huntingtin aggregation and the biomechanics of endocytosis. The methods and data present compelling support of these findings, although there is a need for further additional analyses in this key area.

## Introduction

Aggregation and misfolding of proteins is often linked to a number of neurodegenerative disorders such as Huntington's disease, amyotrophic lateral sclerosis, Alzheimer's disease and Parkinson's disorder (*DiFiglia et al., 1997*; *Chartier-Harlin et al., 1991*; *Bruijn et al., 1997*; *Davies et al., 1997*; *Wilson et al., 2023*).

Huntington's disease (HD), an inherited neurodegenerative disorder, is characterized by the formation of neuronal intracellular inclusions and loss of striatal projection neurons (*Reiner et al., 1988*). The underlying cause of this disease is due to an abnormal expansion of the CAG (coding for the amino acid, Glutamine, Q) tract beyond 35–40 repeats in Exon 1 of the Huntingtin gene (*MacDonald, 1993*), which may promote an abnormal protein conformation, resulting in the formation of aggregates observed in the nucleus and cytoplasm of affected neurons (*Lunkes et al., 1999*). Huntingtin aggregates are capable of spreading from cell – to – cell in the *Drosophila* brain, resulting in a loss of the neuronal population. However, neuronal loss could be prevented by blocking endocytosis in the recipient neurons or synaptic exocytosis (*Babcock and Ganetzky, 2015*). In addition to affecting neurons, mutant forms of Huntingtin also affect other non-neuronal cells (*Sathasivam et al., 1999*; *Weiss et al., 2012*). Abnormalities in the peripheral, non-CNS tissues of HD patients are consistently observed, resulting in weight loss, altered glucose homeostasis, and sub-cellular abnormalities (*Sassone et al., 2009*). Aberrant immune responses are observed in the presence of Huntingtin aggregates (*Nayak et al., 2011*). Expression of Huntingtin aggregates resulted in a compromised immune function in *Drosophila* hemocytes resulting in increased susceptibility to infections, and altered production of cytokines and JAK-Stat pathway activity (*Lin et al., 2019*). An effective immune response is largely dependent on functional endocytosis (*Devergne et al., 2007*; *Huang et al., 2010*). Interestingly, internalization of transferrin, a cargo internalized via clathrin-mediated endocytosis (CME) was drastically reduced in the presence of aggregated forms of Huntingtin and other pathogenic polyQ aggregates, suggestive of altered endocytosis (*Yu et al., 2014*).

CME is a critical process involved in the cellular uptake of various molecules from the extracellular milieu and the plasma membrane. CME plays a critical role in synaptic vesicle trafficking (*Royle and Lagnado, 2010*), with clathrin-coated structures (CCSs) being central to this process (*Conner and Schmid, 2003*). Alterations in CME have also been implicated in neurodegenerative disorders such as Huntington's disease, spinocerebellar ataxia type I and Amyotrophic lateral sclerosis. Previous reports demonstrate that protein aggregates formed by mutant forms of Huntingtin (HTT), Ataxin-1 (Atx1) and Superoxide dismutase-1 (SOD1) can inhibit CME and AMPA receptor recycling (*Yu et al., 2014*). The polyQ containing Huntingtin exon 1 fragment is also known to interact with proteins involved in CME including the Clathrin heavy chain, Huntingtin interacting protein 1 (Hip1), Adaptor protein complex 2 alpha 2 subunit and Dynamin1 (*Kaltenbach et al., 2007*). Further, uptake of transferrin, a ligand that undergoes endocytosis was reduced in the presence of Huntington aggregates, while mutations in genes involved in endocytosis enhanced Huntington-associated toxicity (*Meriin et al., 2003*).

Actin is known to play a central and critical role in the context of CME (*Merrifield et al., 2002*). Alterations in the organization of actin are known to regulate the physical properties of cells (*Mote et al., 2020*), which in turn are shown to impact the health and function of cells. A recent study, using Huntingtin knockout mice demonstrated that loss of Huntingtin resulted in hyperactivation of LIM kinase and stabilized the Actin cytoskeleton in a Cofilin-dependent manner (*Wennagel et al., 2022*).

In the current study we set out to study how CME and actin dynamics were affected in the presence of pathogenic protein aggregates using live cell imaging. We further queried whether a common mechanism operated in all aggregate-induced neurodegenerative conditions. Our study reveals that the movement and directionality of CCSs are affected in the presence of pathogenic Huntingtin aggregates, accompanied by alterations in actin flow in *Drosophila* hemocytes and mammalian cells. We further demonstrate that pathogenic forms of Huntingtin disrupt actin reorganization by sequestering Arp2/3 complex components and actin, with overexpression of either Arp3 or Hip1 partially rescuing the stalled CCS movement phenotype along with restoring their directional movement. Arp3 or Hip1 overexpression also rescued Huntingtin-driven neurodegeneration observed in the *Drosophila* eye. Using atomic force microscopy (AFM), we determined that cells containing Huntingtin aggregates display an increased cellular stiffness in an actin-dependent manner. Interestingly, overexpression of Hip1 or transient treatment with Latrunculin A restored cellular stiffness to levels comparable to wild

type cells. Further, upon testing the effects of other pathogenic aggregates on CME and actin organization and dynamics, we found a strong correlation between disruption of CCS dynamics, actin flow and enhanced cellular stiffness. Together our results indicate that Huntingtin aggregates remodel the cellular actin cytoskeleton in a manner rendering the cells stiffer, where it is unable to assist CCS movement. We further demonstrate that an active remodeling of the actin cytoskeleton can override some of the detrimental effects of the aggregates.

## Results

### Clathrin-mediated endocytosis and clathrin-coated structure dynamics are compromised in the presence of pathogenic Huntingtin polyQ aggregates

Previous reports have demonstrated that *Drosophila melanogaster* expressing mutant forms of the Huntingtin gene display many of the phenotypes observed in affected human patients, including accumulation of protein aggregates in cells and reduced life span (*Weiss et al., 2012*; *Rosas-Arellano et al., 2018*; *Bradford et al., 2010*). Further, non-neuronal *Drosophila* cells also display altered cellular properties in the presence of pathogenic Huntingtin aggregates (*Lin et al., 2019*). Other reports have demonstrated that cells containing Huntingtin aggregates display reduced transferrin uptake indicating that endocytosis may be compromised (*Yu et al., 2014*).

CME is a highly dynamic process (*Mettlen and Danuser, 2014*). We used live cell imaging to better understand how movement of CCSs is affected in cells expressing HTT Q138. This mutant form of the Huntingtin protein contains the first 588 aa of the protein in addition to the polyQ stretch. To track the movement of CCSs in real time, we used a transgenic *Drosophila* line in which clathrin light chain (Clc) is fused to GFP and co-expressed this with pathogenic HTT Q138 or non-pathogenic HTT Q15 or in WT cells. In the WT or non-pathogenic HTT Q15 (data not shown) expressing hemocytes, we observed a centripetal movement of CCSs similar to previous observations (*Kochubey et al., 2006*; *Figure 1A and B*). We found no difference in CCS movement between WT cells and cells expressing HTT Q15, and henceforth use these two cell types interchangeably (*Figure 1A* and data not shown). However, in HTT Q138 expressing cells there was little to no movement of these CCSs (*Figure 1B*; *Figure 1—video 1*, *Figure 1—video 2*). Using particle image velocimetry (PIV) analysis (see Materials and methods for details and *Figure 1—figure supplement 1*) to quantify the movement and directionality of CCSs, we determined that in WT/HTT Q15 expressing cells, CCSs follow a specific path with defined speeds (*Figure 1A′*), whereas CCSs in HTT Q138 expressing cells do not show any directional movement, with negligible radial speed indicating a severe compromise in clathrin dynamics and movement of clathrin-coated structures (*Figure 1B′*). Analysis of the distribution of flow-field directions obtained from PIV analysis relative to the polar direction revealed that the angles were sharply distributed around a value of 180°, showing the centripetal movement of CCSs (*Figure 1A″*). However, in the case of HTT Q138 hemocytes, a broad distribution of the angles was obtained, indicating the absence of any directional centripetal movement of CCSs (*Figure 1B″*). To check whether ligand uptake was also reduced, we determined the ability of hemocytes to internalize maleylated BSA (mBSA), a cargo for the anionic ligand binding scavenger receptor. HTT Q138 expressing hemocytes showed reduced colocalization of mBSA with CCSs coupled with reduced internalization in comparison to control hemocytes (*Figure 1C and D*). We also determined the movement of CCSs in mammalian cells, HEK293T and SH-SY5Y expressing either HTT Q15 or HTT Q138. We observed a loss in movement of CCSs exclusively in the presence of HTT Q138 aggregates, similar to what was observed in hemocytes (*Figure 1—figure supplement 2*), indicating the existence of similar deficits across cell types and organisms.

### Presence of Huntingtin polyQ aggregates negatively affect actin dynamics

We next addressed what might be a plausible explanation for the stagnation in CCS movement in HTT Q138 expressing cells. We hypothesized that this might be due to malfunction associated with cytoskeletal elements. Previously using inhibitors such as Latrunculin A, colchicine or nocodazole, it was shown that the centripetal CCS movement in hemocytes was dependent on the actin cytoskeleton and not on microtubules (*Kochubey et al., 2006*). The role of the actin cytoskeleton is also well established

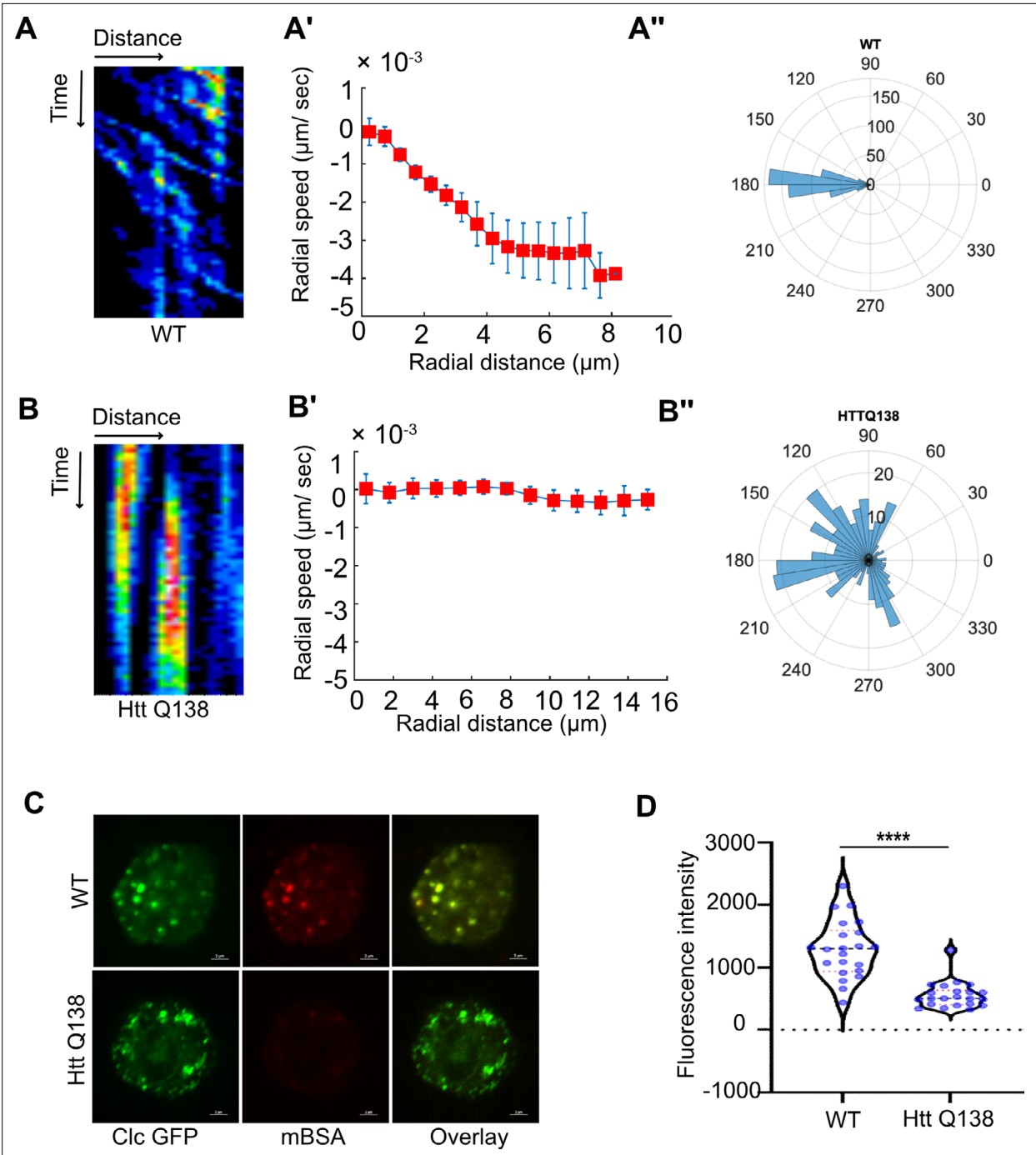

**Figure 1.** Clathrin mediated endocytosis and clathrin coated structure dynamics are compromised in the presence of pathogenic Huntingtin polyQ aggregates. (**A**) Kymograph showing movement of CCSs by live cell imaging of clathrin light chain tagged with GFP in WT hemocytes. X-axis represents distance and Y-axis represents time. (**A'**) Radial speed (µm/ sec) of the CCSs in WT hemocytes as a function of radial distance (in µm) from the cell center obtained from time-averaged PIV analysis (see Materials and methods for details). (**A''**) Polar histogram of distribution of the flow-field directions obtained from PIV analysis relative to the polar direction (see Materials and methods for details). The angles are sharply distributed around a value of 180°, showing the centripetal movement of CCSs. (**B**) Kymograph showing stalled movement of CCSs by live cell imaging of clathrin light chain tagged with GFP in HTT Q138 hemocytes. (**B'**) Graph showing radial speed (µm/ sec) of the CCSs in HTT Q138 hemocytes. (**B''**) Polar histogram of flow-field directions obtained similar to the WT case in A'' gives a broad distribution of the angles, indicating the absence of any directional centripetal movement of CCSs in presence of HTT Q138. (**C**) Internalization of mBSA in WT and HTT Q138 cells. Clathrin puncta are shown in green and mBSA in the red channel. Scale bar 2 µm. (**D**) Graph shows the quantification of internalized mBSA. Y-axis represents the intensity of mBSA internalized under different conditions. N=20 cells. p-value <0.0001 (Mann-Whitney test).

*Figure 1 continued on next page*

*Figure 1 continued*

The online version of this article includes the following video and figure supplement(s) for figure 1:

**Figure supplement 1.** Time-averaged CCS flow-field obtained from the PIV analysis for wild type cells.

**Figure supplement 2.** CCS movement is stalled in the presence of pathogenic HTT aggregates in mammalian cells.

**Figure 1—video 1.** Video showing time lapse imaging of a HTT Q15 hemocyte expressing clathrin light chain tagged with GFP over 5 mins at 5 s intervals.

https://elifesciences.org/articles/98363/figures#fig1video1

**Figure 1—video 2.** Video showing time lapse imaging of an HTT Q138 hemocyte expressing clathrin light chain tagged with GFP over 5 min at 5 s intervals.

https://elifesciences.org/articles/98363/figures#fig1video2

in the context of CME and CCS movement across various mammalian cell lines and in yeast (*Meriin et al., 2003*; *Yarar et al., 2005*; *Ferguson et al., 2009*). Similar to what was previously reported, we found that hemocytes treated with pharmacological inhibitors of actin polymerization, Latrunculin A (LatA) or Cytochalasin D (CytoD) showed severe retardation in CCS movement in comparison to their DMSO treated counterparts (*Figure 2A and B* and *Figure 2—figure supplement 1*). This was also accompanied by a loss of CCSs speed, directionality and centripetal movement (*Figure 2A', A'', B' and B''*). As expected, actin dynamics in hemocytes treated with either Lat A or CytoD was also affected (data not shown). A reduction in the number of filopodia in Lat A and Cyto-D-treated cells was also observed in comparison to DMSO-treated cells (*Figure 2—figure supplement 1*).

To further understand how HTT Q138 impacted actin function, we co-expressed Actin-GFP along with HTT Q138. We observed sequestration of Actin-GFP in the HTT aggregates (*Figure 2—figure supplement 1*). As this sequestration may result in removal and reduction of functional and available actin in the cells, we reasoned this might affect the overall actin dynamics in HTT Q138 expressing cells. We performed live cell imaging of hemocytes co-expressing Lifeact-GFP in either WT/HTT Q15 or HTT Q138 expressing cells. While we observed a dynamic actin cytoskeleton accompanied by numerous filopodia extensions and retractions in WT/HTT Q15 expressing cells, there was a dramatic alteration in the dynamics of actin and reduced filopodia number and length in HTT Q138 expressing cells (*Figure 2C, D and E*, *Figure 2—figure supplement 1* and *Figure 2—video 1* and *Figure 2—video 2*). Measurement of cortical actin revealed that the thickness as well as intensity was significantly increased in presence of HTT Q138 compared to HTT Q15 expressing cells (*Figure 2F and G*). The reduction in filopodia formation and increase in presence of cortical actin in the presence of HTT Q138 suggests an altered organization of actin. To study the polymerization of actin in HTT Q15/Q138 expressing cells, we performed barbed end labelling of actin by providing fluorescently tagged actin, similar to what has been previously described (*Pardee and Spudich, 1982*). A significant reduction of incorporated actin was observed in HTT Q138 expressing cells (*Figure 2H I*), indicating that polymerization of actin was hampered in the presence of HTT Q138. This was further supported by a reduction in actin flow in hemocytes expressing HTT Q138 compared to WT cells (*Figure 2J*).

Formation of filopodia requires the activity of the Arp2/3 complex (*Korobova and Svitkina, 2008*). Mass spectrometric analysis of Huntingtin aggregates formed in the brain indicated a sequestration of a number of proteins involved in actin remodeling and endocytosis. Among these were components of the Arp2/3 complex, additional actin-modulating proteins, clathrin heavy and light chains and Hip1 and Hip1r, proteins that are known to connect clathrin pits to actin at sites of endocytosis, although the nature of their sequestration and association with aggregates remains unknown (*Hosp et al., 2017*; *Kim et al., 2016*). Similar to what was previously demonstrated, we found significant enrichment of Arp3, a key component of the Arp2/3 complex within HTT Q138 aggregates (*Figure 3A*) indicating atleast a partial sequestration of one of the components of the Arp2/3 complex. We speculated that the sequestration of specific components of the Arp2/3 complex may alter the stoichiometry, thus potentially rendering the complex non-functional in the presence of HTT Q138 aggregates. We further reasoned that we should see a similar impairment of CCS movement upon knocking down Arp3 directly in hemocytes even in the absence of HTT Q138. Upon inducing expression of a transgenic Arp3 RNAi construct in hemocytes, we observed a reduction in the number of filopodia (*Figure 3B* and *Figure 3—figure supplement 1*) along with stalled movement of CCSs (*Figure 3C and C'*) and

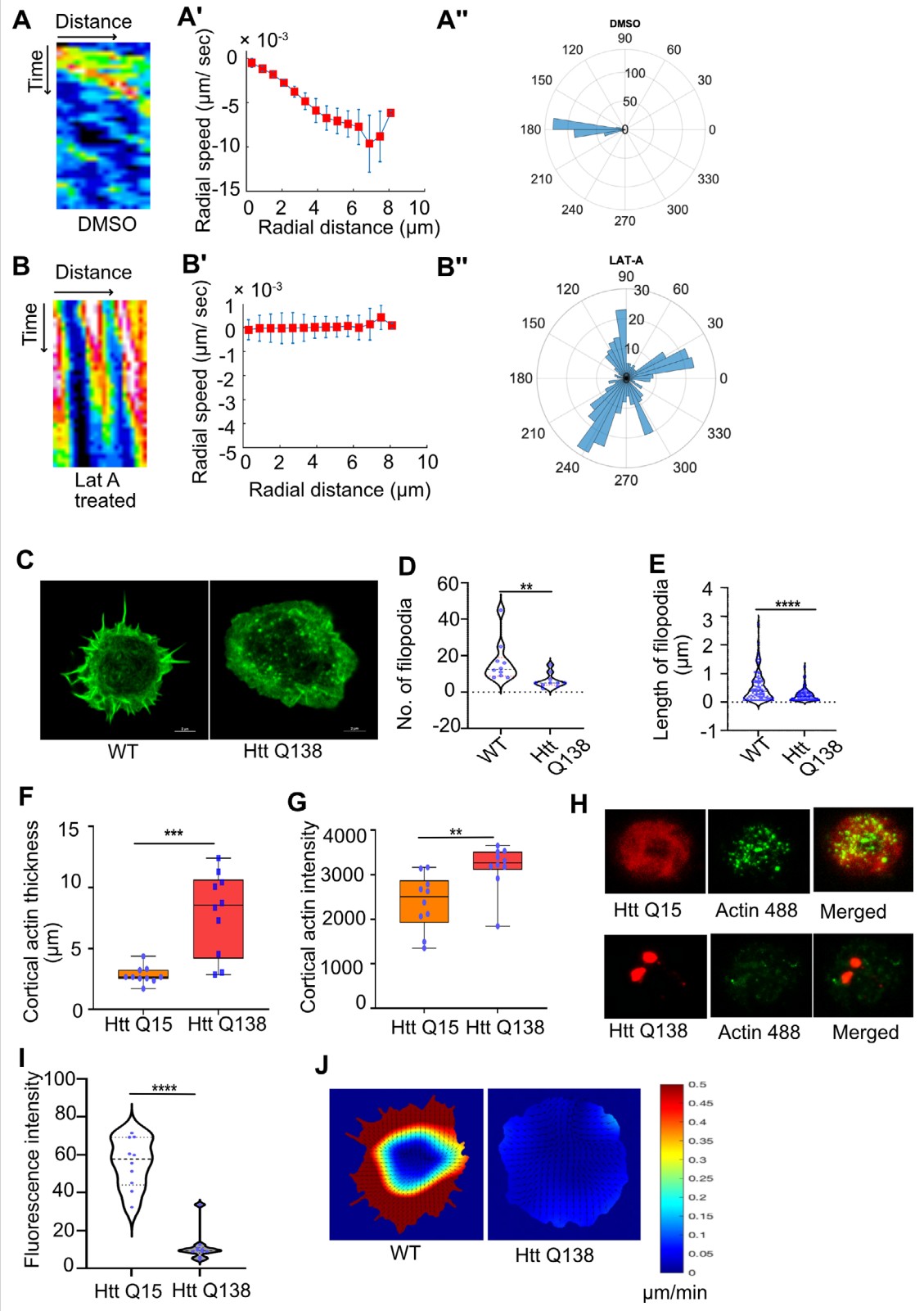

**Figure 2.** Presence of Huntingtin polyQ aggregates negatively affect Actin dynamics. (**A**) Kymographs showing movement of CCSs by live cell imaging of clathrin light chain tagged with GFP in DMSO treated cells. (**A'**) Radial speed (μm/ s) of the CCSs as a function of radial distance (in μm) from the cell center obtained from time-averaged PIV analysis. (**A''**) Polar histogram of distribution of the flow-field directions obtained from PIV analysis relative to the polar direction (see Materials and methods for details). The angles are sharply distributed around a value of 180°, showing the centripetal movement

*Figure 2 continued on next page*

*Figure 2 continued*

of CCSs. (**B**) Kymographs showing movement of CCSs by live cell imaging of clathrin light chain tagged with GFP in LatA-treated cells. (**B'**) Radial speed (μm/ sec) of the CCSs as a function of radial distance (in μm) from the cell center obtained from time-averaged PIV analysis shows stalled movement of CCSs (**B"**). Polar histogram of flow-field directions gives a broad distribution of angles, indicating the absence of any directional centripetal movement of CCSs upon LatA treatment. (**C**) Representative micrograph showing filopodia formation in HTT Q15, whereas no filopodia are seen in HTT Q138 expressing cells. Scale bar 5 μm. (**D**) Graphs showing the quantification of the number of filopodia in WT and HTT Q138 expressing cells. Number of cells = 10, p-value = 0.001 and (**E**) shows quantification of length of filopodia in WT and HTT Q138 expressing cells. Number of cells = 10, p-value <0.0001 (Mann-Whitney test). (**F, G**) Box plot showing increase in cortical thickness and actin intensity in cortical region of cells respectively. Number of cells = 10. p-value = 0.0015, p-value = 0.0003 (Mann-Whitney test). (**H**) Micrograph showing barbed-end labelling of actin in hemocytes expressing either HTT Q15 or HTT Q138. HTT Q15/Q138 are shown in red, whereas barbed ends labeled with actin are shown in the green channel. (**I**) Violin plot shows quantification of labeled barbed end in HTT Q15 and HTT Q138 expressing cells. Number of cells = 10. p-value <0.0001 (Mann-Whitney test). (**J**) Representative images showing actin flow in wild type and HTT Q138 expressing cells, obtained from PIV analysis.

The online version of this article includes the following video and figure supplement(s) for figure 2:

**Figure supplement 1.** Presence of Huntingtin polyQ aggregates alters Actin dynamics.

**Figure 2—video 1.** Video showing time lapse imaging of a WT hemocyte expressing Lifeact tagged with GFP over 5 min at 5 s intervals.
https://elifesciences.org/articles/98363/figures#fig2video1

**Figure 2—video 2.** Video showing time lapse imaging of a hemocyte expressing Lifeact tagged with GFP in presence of HTT Q138 over 5 min at 5 s intervals.
https://elifesciences.org/articles/98363/figures#fig2video2

loss of directional movement (*Figure 3C"*). Similar results were also obtained upon treatment of wild type cells with CK666, a chemical inhibitor of Arp2/3 (*Figure 3—figure supplement 1*).

We also disrupted actin polymerization genetically by targeting Profilin using RNAi. Porfilin is an actin monomer binding protein and a key regulator of actin polymerization (*Theriot and Mitchison, 1993*). Knockdown of Profilin in hemocytes also resulted in an impairment in CCS movement as well as actin dynamics (*Figure 3D* and *Figure 3—figure supplement 1*). PIV analysis of CCSs upon Profilin knockdown also demonstrated defects in CCS movement and loss of their directionality (*Figure 3D' and D"*). Chemical inhibition of Formin, a Rho GTPase involved in regulation of elongation of actin fibers by SMIFH2, also resulted in stalled CCS movement (*Figure 3—figure supplement 1*), further establishing that remodeling of the actin cytoskeleton is required for directional movement of CCSs.

Since, we found the impairment of CCS movement in HTT Q138 cells to be very similar to the phenotype that we observed on targeting the actin polymerization pathways, we asked if the nature of individual CCSs in both these cases were also similar in terms of clathrin light chain exchange. Towards this we used fluorescence recovery after photo-bleaching (FRAP) to determine the dynamicity of Clathrin light chain exchange in the CCSs in the presence of pathogenic HTT Q138 and compared this to conditions where actin polymerization was perturbed. Recovery of fluorescence intensity of Clc-GFP was severely compromised in the presence of HTT Q138 (*Figure 3—figure supplement 1*) compared to an almost complete recovery in wild type and control (Luc VAL 10) cells. Incomplete recovery was also observed upon knocking down Profilin or upon treatment with Lat A (*Figure 3—figure supplement 1*). This indicates that targeting actin polymerization pathways can mimic the intracellular condition of the presence of pathogenic HTT aggregates by altering the exchange of clathrin light chains at CCSs.

We also examined the involvement of microtubules in the context of HTT Q138. While microtubules are not generally considered to play a role in the early steps of Clathrin mediated endocytosis and clathrin coated vesicle dynamics (*Kochubey et al., 2006*), other studies have demonstrated their involvement in receptor-mediated endocytosis (*Subtil and Dautry-Varsat, 1997*; *Rappoport et al., 2003*; *Popova and Rasenick, 2004*). We found that the organization and dynamics of microtubules remained unchanged even in the presence of Huntingtin aggregates (*Figure 3—videos 1 and 2*) supporting the previous finding (*Kochubey et al., 2006*) that microtubules do not play a role in early events of CME.

We further speculated that the CCS movement on actin cytoskeleton may involve specific motor proteins. Huntingtin is known to interact with Myosin VI via its optineurin binding domain (*Caviston and Holzbaur, 2009*). The actin-based cytoskeleton motor, Myosin VI, is also implicated in various steps of membrane trafficking *Buss et al., 2001b* and shown to be involved in trafficking of clathrin-coated vesicles (*Soldati and Schliwa, 2006*). The co-localization of myosin VI with actin polymerization

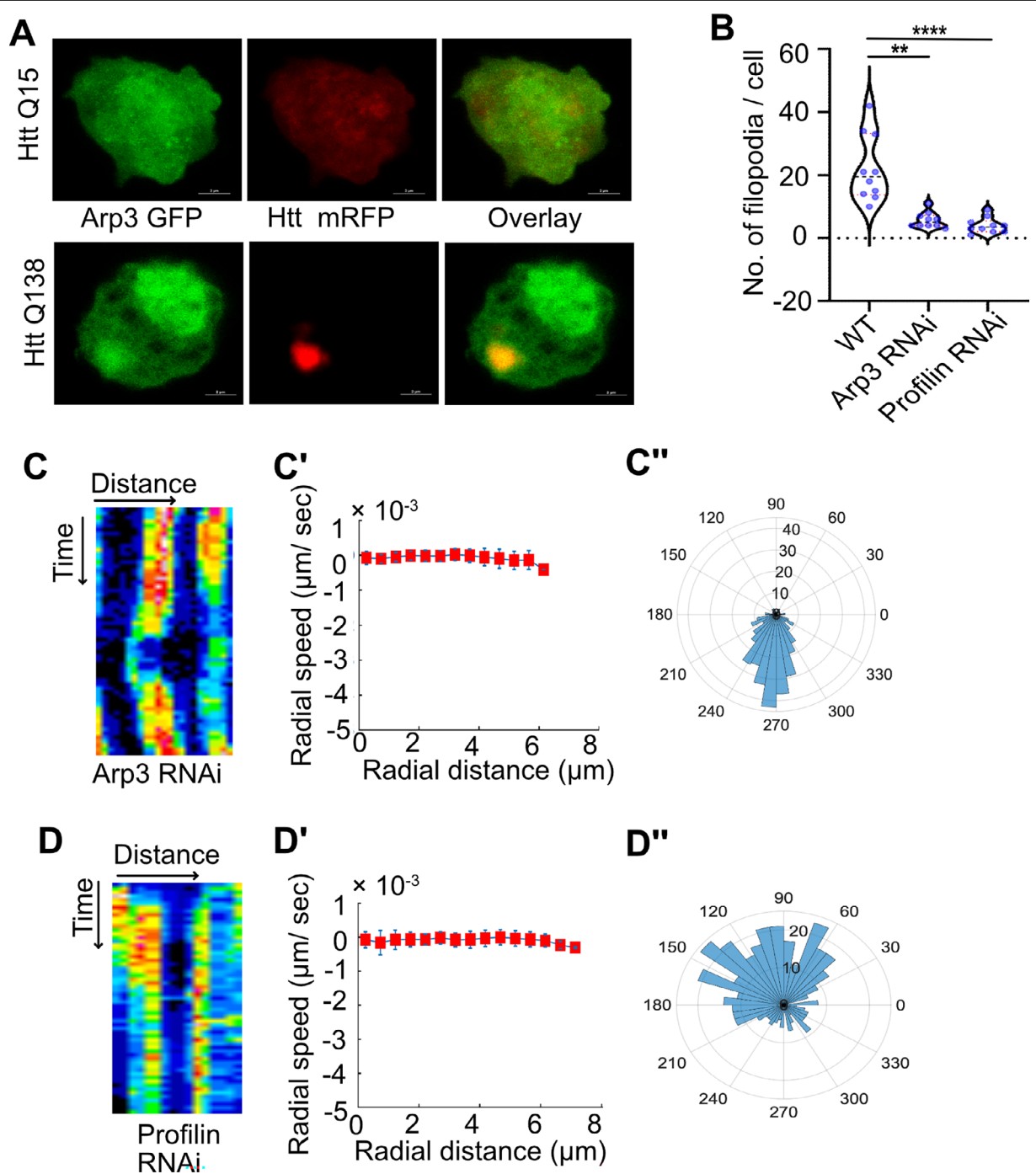

**Figure 3.** Knockdown of components of the Arp2/3 complex or Profilin result in disruption of CCS movement. (**A**) Representative micrographs showing sequestration of Arp3 in HTT Q138 aggregates. Arp3 is shown in green and HTT Q15 /HTT Q138 - RFP is shown in red channel. (**B**) Graph showing quantification of number of filopodia upon Arp3 or Profilin knockdown. Number of cells = 10. p-value = 0.0025, p-value <0.0001 (Kruskal-Wallis test followed by post hoc Dunn's multiple comparisons test). (**C**) Kymograph showing stalled movement of CCSs by live cell imaging of clathrin light chain tagged with GFP upon Arp3 knockdown. (**C'**) Radial speed (μm/ s) of the CCSs as a function of radial distance (in μm) from the cell center obtained from time-averaged PIV analysis shows compromised movement of CCSs upon Arp3 knockdown. (**C"**) Polar histogram of flow-field directions gives a broad distribution of the angles, indicating the absence of any directional centripetal movement of CCSs upon Arp3 knockdown. (**D**) Kymographs showing stalled movement of CCSs by live cell imaging of clathrin light chain tagged with GFP upon Profilin knockdown. (**D', D"**) PIV analysis shows stalled CCS movement and loss in directionality under Profilin knockdown conditions.

The online version of this article includes the following video and figure supplement(s) for figure 3:

*Figure 3 continued on next page*

*Figure 3 continued*

**Figure supplement 1.** CCS movement is disrupted on blocking actin redmodeling.

**Figure 3—video 1.** Video showing time lapse imaging of a WT hemocyte expressing Tubulin tagged with GFP over 5 min at 5 s intervals.

https://elifesciences.org/articles/98363/figures#fig3video1

**Figure 3—video 2.** Video showing time lapse imaging of a hemocyte containing HTT Q138 aggregates expressing Tubulin tagged with GFP over 5 min at 5 s intervals.

https://elifesciences.org/articles/98363/figures#fig3video2

**Figure 3—video 3.** Video showing time lapse imaging of a hemocyte expressing Clathrin light chain tagged with GFP over 5 min at 5 s intervals upon myosin VI knockdown condition.

https://elifesciences.org/articles/98363/figures#fig3video3

**Figure 3—video 4.** Video showing time lapse imaging of a hemocyte expressing Lifeact tagged with GFP over 5 min at 5 s intervals upon myosin VI knockdown condition.

https://elifesciences.org/articles/98363/figures#fig3video4

regulatory proteins such as Cortactin and the Arp2/3 complex has also been shown suggesting a role for this motor in the regulation of actin dynamics. It has also been shown that Myosin VI is associated with CCSs through its C –terminal tail *Buss et al., 2001a*. The coordinated action of myosin VI and Clathrin light chain a (CLCa) are essential for the fission of clathrin-coated pits (*Biancospino et al., 2019*). Knocking down Myosin VI, resulted in an impairment of CCS movement (*Figure 3—figure supplement 1* and *Figure 3—video 3*). However there was no discernible effect on actin dynamics or on filopodia formation (*Figure 3—video 4*).

## Huntingtin-induced CCS movement defect and neurodegeneration can be rescued by actin regulating proteins or chaperones

In order to determine whether the defective CCS movement observed in the presence of pathogenic HTT aggregates was predominantly driven by an alteration in actin dynamics due to sequestration of actin binding proteins by HTT aggregates, we asked whether overexpression of actin binding proteins could rescue the directional movement of CCSs. Towards this we co-expressed Arp3 along with pathogenic HTT Q138 in hemocytes and used live cell imaging to check the status of CCS movement and actin dynamics in these cells. We observed a partial restoration of CCS movement and directionality along with filopodia formation (*Figure 4A, A', A'' and 4E*, *Figure 4—figure supplement 1* and *Figure 4—video 1* and *Figure 4—video 2*).

Huntingtin interacting protein1 (Hip1) is one of several proteins involved in the initial stages of Clathrin mediated endocytosis (*Metzler et al., 2001*; *Engqvist-Goldstein et al., 2001*). Hip1 is also a known interactor of Huntingtin protein and expansions in the polyglutamate tract, as seen in pathological conditions, are known to disrupt this interaction (*Kalchman et al., 1997*). Additionally, Hip1 is also known to interact with actin *Senetar et al., 2004* and the Hip1-actin interaction is known to be regulated by the clathrin light chain (*Wilbur et al., 2008*). Further, as previously mentioned, Hip1 was also found to be sequestered in pathogenic Huntingtin aggregates (*Hosp et al., 2017*; *Qin et al., 2004*). These studies indicate that the normal function of Hip1 of interacting with actin and clathrin is disrupted in the context of Huntington's disorder. Upon co-expressing Hip1 with pathogenic HTT Q138, we observed a partial rescue of CCS movement (*Figure 4B, B' and B''*, and *Figure 4—video 3*). Conversely, knocking down Hip1 in wild type hemocytes resulted in a reduction in movement (*Figure 4C*), speed, and loss of directionality of CCSs (*Figure 4C' and C''*), thus highlighting the importance of the Hip1- actin axis in CME and its possible disruption in the presence of HTT Q138.

HTT Q138 forms aggregates inside the cell due to misfolding of protein, and its toxicity and aggregation was reported to be modulated by DNAJ chaperones (*Labbadia et al., 2012*; *Warrick et al., 1999*). Previously, *Drosophila* Mrj, a homologue of mammalian DnaJB6 was observed to rescue polyglutamine associated toxicity in *Drosophila*. In a screen for polyQ modifiers, DNAJB6, a member of the DNAJ (HSP40) chaperone family was found to efficiently suppress polyQ aggregation in cells (*Hageman et al., 2010*). Previous studies in mice and *Drosophila* have also shown that polyglutamine toxicity and aggregation can be modulated by DNAJ chaperones (*Labbadia et al., 2012*; *Warrick et al., 1999*). The DNAJ domain containing protein Mrj, which is highly enriched in the brain, can effectively suppress polyQ toxicity (*Chuang et al., 2002*). Additionally, Mrj was also identified in an

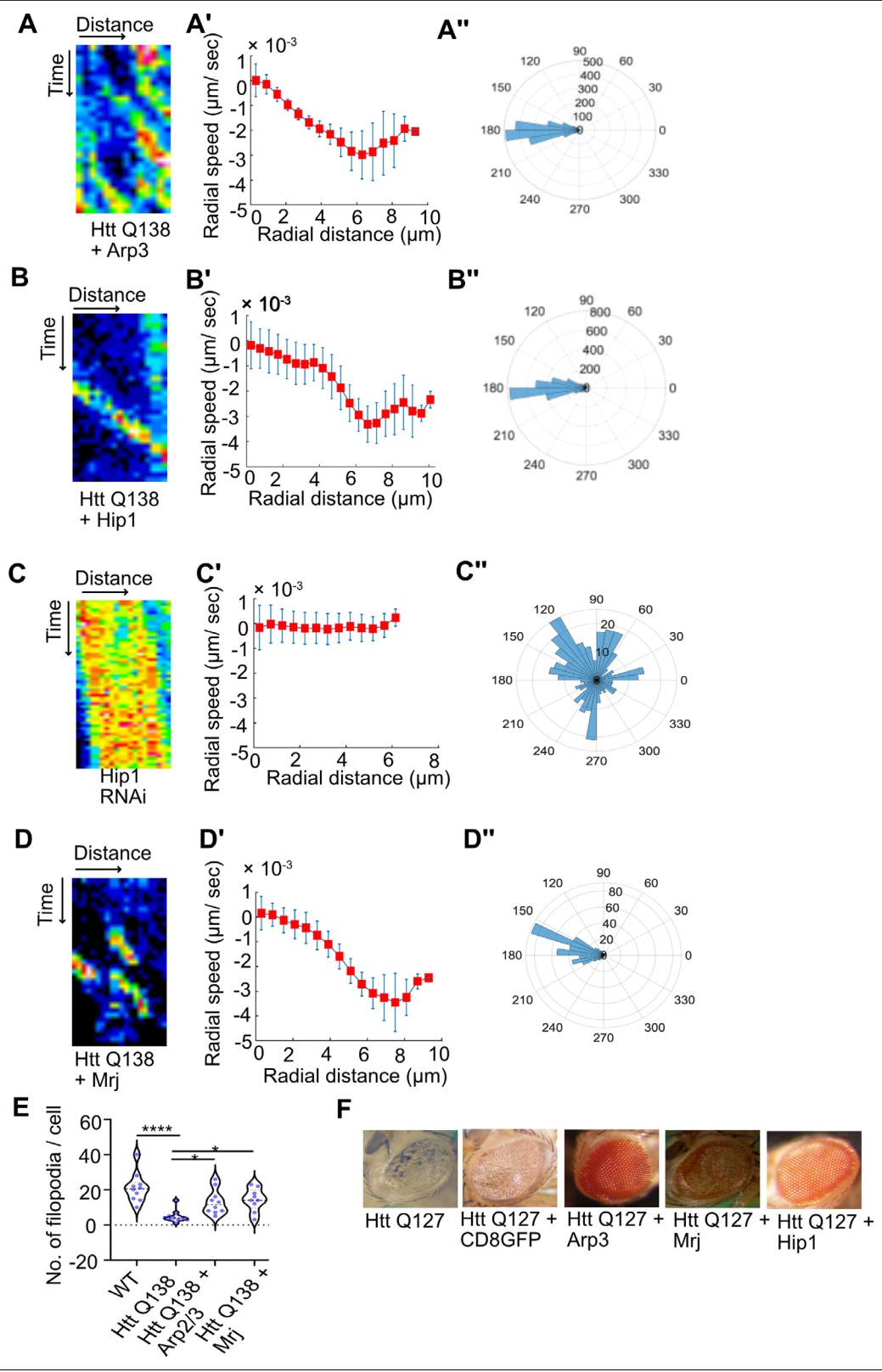

**Figure 4.** Overexpression of components of the Arp2/3 complex or Hip1 rescue CCS movement and neurodegeneration even in the presence of pathogenic HTT. (**A**) Kymograph showing movement of CCSs by live cell imaging of clathrin light chain tagged with GFP in HTT Q138 +Arp3 condition. (**A'**) Radial speed (μm/ s) of the CCSs as a function of radial distance (in μm) from the cell center obtained from time-averaged PIV analysis. (**A''**)

*Figure 4 continued on next page*

*Figure 4 continued*

Polar histogram of distribution of the flow-field directions obtained from PIV analysis relative to the polar direction are shown. The angles are sharply distributed around a value of 180°, showing the centripetal movement of CCSs. (**B**) Kymograph showing movement of CCSs by live cell imaging of clathrin light chain tagged with GFP in HTT Q138 +Hip1 condition. (**B'**) Radial speed (µm/ s) of the CCSs as a function of radial distance (in µm) from the cell center obtained from time-averaged PIV analysis. (**B''**) Polar histogram of distribution of the flow-field directions obtained from PIV analysis relative to the polar direction. The angles are sharply distributed around a value of 180°, showing the centripetal movement of CCSs in HTT Q138 +Hip1 condition. (**C**) Kymograph showing stalled movement of CCSs by live cell imaging of clathrin light chain tagged with GFP in Hip1 knockdown cells. (**C'**) Radial speed (µm/ s) of the CCSs as a function of radial distance (in µm) from the cell center obtained from time-averaged PIV analysis shows compromised movement of CCSs in Hip1 knockdown cells. (**C''**) Polar histogram of flow-field directions gives a broad distribution of the angles, indicating the absence of any directional centripetal movement of CCSs upon Hip1 knockdown. (**D**) Kymograph showing movement of CCSs by live cell imaging of clathrin light chain tagged with GFP in HTT Q138 +Mrj cells. (**D'**) Radial speed (µm/ s) of the CCSs as a function of radial distance (in µm) from the cell center obtained from time-averaged PIV analysis in HTT Q138 +Mrj cells. (**D''**) Polar histogram of distribution of the flow-field directions obtained from PIV analysis relative to the polar direction shows the angles are sharply distributed around a value of 180°, showing the centripetal movement of CCSs. (**E**) Violin plot showing quantification of number of filopodia in WT, HTT Q138, HTT Q138 +Arp3 and HTT Q138 +Mrj cells. Number of cells = 10, p-value = 0.0309 for HTT Q138 +Mrj and p-value = 0.0154 for HTT Q138 +Arp3 compared to HTT Q138, p-value <0.0001 (Kruskal-Wallis test followed by post hoc Dunn's multiple comparisons test). (**G**) Representative micrograph showing *Drosophila* eye expressing HTT Q127, HTT Q127 +CD8 GFP, HTT Q127 +Arp3, HTT Q127 +Mrj and HTT Q127 +Hip1.

The online version of this article includes the following video and figure supplement(s) for figure 4:

**Figure supplement 1.** Micrograph showing representative images obtained from filoquant for quantification of number and length of filopodia upon overexpression of Arp3 or Mrj in the HTT Q138 background.

**Figure 4—video 1.** Video showing time lapse imaging of a hemocyte expressing Clathrin light chain tagged with GFP over 5 min at 5 s intervals under HTT Q138 +Arp2/3 condition.

https://elifesciences.org/articles/98363/figures#fig4video1

**Figure 4—video 2.** Video showing time lapse imaging of hemocyte expressing Lifeact tagged with GFP over 5 min at 5 s intervals under HTT Q138 +Arp2/3 condition.

https://elifesciences.org/articles/98363/figures#fig4video2

**Figure 4—video 3.** Video showing time lapse imaging of a hemocyte expressing Clathrin light chain tagged with GFP over 5 min at 5 s intervals under HTT Q138 +Hip1 condition.

https://elifesciences.org/articles/98363/figures#fig4video3

**Figure 4—video 4.** Video showing time lapse imaging of a hemocyte expressing Clathrin light chain tagged with GFP over 5 min at 5 s intervals under HTT Q138 +Mrj condition.

https://elifesciences.org/articles/98363/figures#fig4video4

---

independent screen to modulate the cytotoxicity in the context of protein aggregation (*Desai et al., 2022*). We found that coexpressing Mrj along with HTT Q138 in hemocytes partially restored CCS movement (*Figure 4D*) speed and directionality (*Figure 4D' and D''* and *Figure 4—video 4*) along with filopodia formation (*Figure 4E* and *Figure 4—figure supplement 1*). Additionally, overexpression of either Arp3, Hip1 or Mrj could also rescue the neurodegeneration observed in *Drosophila* eyes caused by the overexpression of pathogenic HTT Q127 (*Figure 4F*). This indicates that mechanisms resulting in actin reorganization may contribute largely to the neurodegeneration observed in Huntington's disorder. Further, increasing the availability of proteins involved in actin reorganization are capable of restoring CME even in the presence of pathogenic aggregates. The interplay between actin, CME and pathogenic Huntingtin appears to be largely responsible for the toxicity and intracellular effects observed in the presence of mutant forms of Huntingtin.

## HTT Q138 expressing cells show increased cellular stiffness

A handful of previous studies have focused on studying the physical properties and transition of soluble oligomers into insoluble fibrils. In the past, atomic force microscopy (AFM) has been used to measure the kinetics of aggregate formation (*Ruggeri et al., 2019*). However, these have been carried out using purified proteins, and how these aggregates affect the physical state of a cell in which they are present, remains an unanswered question. Using AFM, we performed nanoindentation on wild type

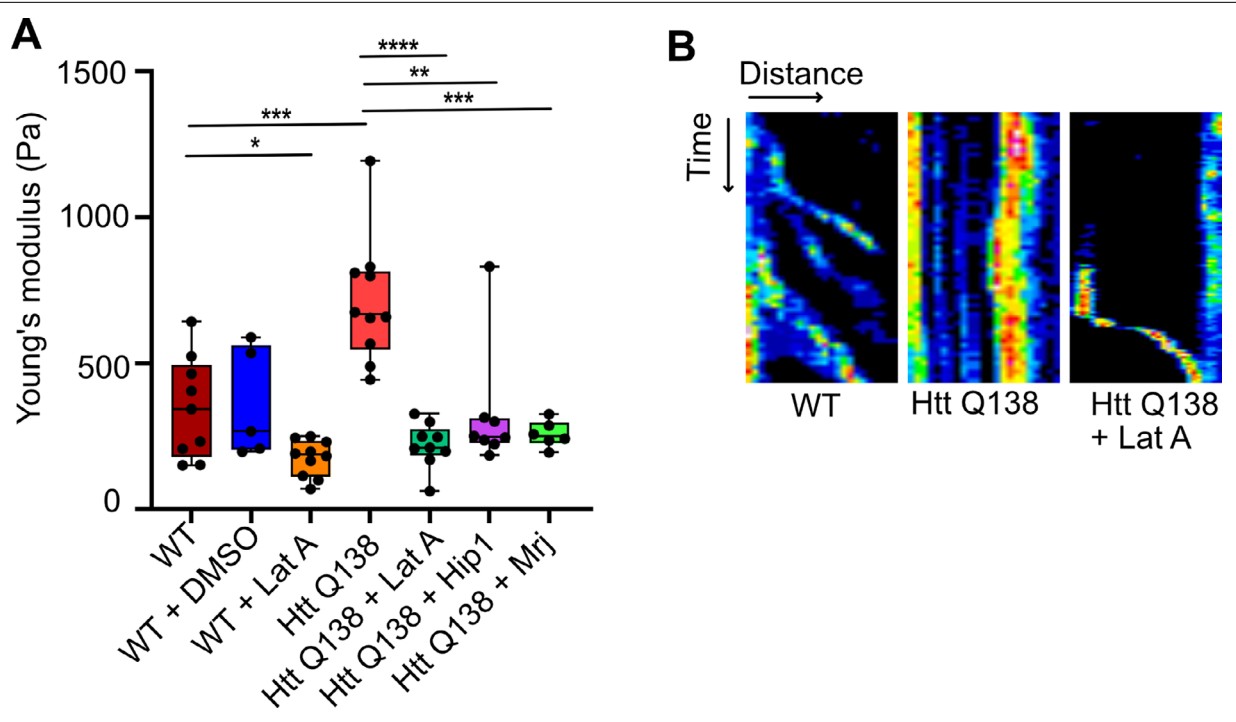

**Figure 5.** HTT Q138 expressing cells show increased cellular stiffness. (**A**) Box plot showing Young's modulus of WT, DMSO, WT +Lat A, HTT Q138, HTT Q138 +Lat A, HTT Q138 +Hip1 and HTT Q138 +Mrj cells. Number of cells: WT = 9; WT +DMSO = 5; HTT Q138=10; WT +Lat A=10; HTT Q138 +Lat A=9; HTT Q138 +Mrj = 6; HTT Q138 +Hip1=7. p-value = 0.0350, p-value = 0.0044 p-value <0.005, ****, p-value <0.0001 (Mann-Whitney test). (**B**) Kymographs showing movement of CCSs by live cell imaging of clathrin light chain tagged with GFP upon transient treatment with Lat A.

The online version of this article includes the following figure supplement(s) for figure 5:

**Figure supplement 1.** A representative force curve obtained on wild type cells.

**Figure supplement 2.** Representative force curves of each cell type fitted to Hertz's model.

**Figure supplement 3.** Box plot showing reduction in stiffness of cells upon treatment with CK666, *P*-value <0.0001 (Mann-Whitney test).

and HTT Q138 expressing cells to measure their Young's modulus (or stiffness). The bead-attached cantilever was approached towards a single cell with a constant speed of 2 µm/s. After contact with the cell, the cantilever was pressed further to deform the cells by ~500 nm and retracted back with the same speed. In the complete approach-retract cycle, the cantilever bending was recorded at every point of cantilever displacement which was then converted to force vs cell–deformation curve known as the force-curve. A representative force-curve on a wild type cell is shown in *Figure 5—figure supplement 1*. Here, the points (a), (b), (c), (d), and (e) represent the conditions at various phases of cell deformations while the bead indents on the cell, with the bead leaving the cell surface at point (e), compared to the point at which it makes contact while approaching (a). *Figure 5—figure supplement 2* shows representative force curves from each cell type. We analyzed the force-curves by fitting them to the Hertz model. Our results demonstrated that cells expressing HTT Q138 were stiffer than WT cells, as indicated by the higher Young's modulus (or stiffness; *Figure 5A*). The stiffness of cells expressing pathogenic Huntingtin aggregates was partially rescued upon overexpression of either Hip1 or Mrj (*Figure 5A*) indicating that partial restoration in actin organization may also rescue stiffness of these cells in addition to CME. We observed reduced CCS movement and reduction in filopodia formation in Arp3 knockdown condition (*Figure 3C, C', C", and B*) and conversely, overexpression of Arp3 in the HTT Q138 background could rescue the CCS movement and filopodia formation (*Figure 4A, A', A", and E*). Further, HTT Q138 cells had increased cortical actin (*Figure 2F*). We performed AFM on cells treated with CK666, a inhibitor of the Arp2/3 complex and found that treatment of CK666 led to a decrease in stiffness of cells compared to DMSO-treated cells (*Figure 5—figure supplement 3*). We also noticed a decreased stiffness of cells upon LatA treatment in comparison to DMSO-treated cells (*Figure 5A*) suggesting that an altered state of the cytoskeleton was largely responsible for

contributing to the increased stiffness observed in the presence of HTT Q138. Our results suggest that due to the increased stiffness of HTT Q138 cells CCS movement may be impaired. Conversely, cells treated with LatA, in which the cytoskeletal architecture is completely broken down, are too soft and are hence also unable to support CCS movement. We therefore hypothesized that a brief, transient treatment of Lat A to HTT Q138 cells may result in a partial movement of CCSs at an intermediate time point where the stiff actin cytoskeleton is being converted to a more dynamic form. We were able to see movement of CCSs in these cells at an intermediate time point (*Figure 5B*) thereby strengthening the idea that CCS movement requires a dynamic form of the actin cytoskeleton and that HTT Q138 causes the cytoskeleton to be held in a very stiff state. A more careful analysis of data, may reveal mechanisms by which stiffness is altered and its correlation with transport properties.

## Screening with other aggregating proteins associated with neurodegeneration show deficits in CCS movement and Actin dynamics in the presence of pathogenic TDP-43

As the presence of misfolded protein aggregates is also a hallmark of several other neurodegenerative conditions, we performed a systematic analysis of CCS movement and Actin dynamics in the presence of other such proteins. Towards this we co-expressed Aβ–42–2 x, FUSR521C, αSynA30P, αSynA53T, TDP-43 along with Clc-GFP and used live imaging to study CCS dynamics in *Drosophila* hemocytes. We observed that while Aβ–42–2 x, FUSR521C (FUS), αSynA30P or αSynA53T (*Figure 6A* and *Figure 6—video 1–4*) caused no defect in CCS movement, in the presence of TDP-43 we noticed a significant reduction in CCS movement (*Figure 6A and B* and *Figure 6—video 5*). Tracking and quantification of CCSs in WT cells and cells expressing mutant proteins revealed that CCSs moved inwards in all cases, except TDP-43 expressing cells, in which movement was completely stalled (*Figure 6A and B* and *Figure 6—video 5*), similar to what we observed in the presence of HTT Q138. This was also supported by analysis of the mean instantaneous velocities, which showed a significant decrease in the presence of TDP-43 (*Figure 6C*). Additionally, the uptake of fluorescently tagged mBSA was markedly reduced in the presence of TDP-43 (*Figure 6D*) indicative of a reduction in CME. In order to determine whether the recruitment or exchange of clathrin to CCSs was altered in the presence of aggregating pathogenic proteins, we performed FRAP on individual CCSs in hemocytes. Recovery of clathrin was severely compromised only in the presence of TDP-43, similar to HTT Q138, and was indistinguishable from WT in the context of other aggregates (*Figure 6—figure supplement 1*).

We also looked at Actin dynamics using LifeAct-GFP in the presence of these pathogenic proteins. Here again, we saw no significant difference in filopodia formation in the hemocytes expressing Aβ–42–2 x (*Figure 7A*, *Figure 7—figure supplement 1*, *Figure 7—video 1*), FUSR521C (*Figure 7A*, *Figure 7—figure supplement 1*, *Figure 7—video 2*) or both mutants of α-Synuclein (*Figure 7A*, *Figure 7—figure supplement 1*, *Figure 7—videos 3 and 4*), but noticed a dramatic reduction in the presence of TDP–43 (*Figure 7A*, *Figure 7—figure supplement 1*, and *Figure 7—video 5*). Number, length and lifetime of filopodia were also compromised in the presence of TDP-43 expressing cells as compared to wild type cells (*Figure 7B, C and D*), while remaining largely unaffected in the context of other mutants. Reduced actin flow was also observed in TDP-43 expressing cells in comparison to WT cells and cells expressing other aggregates (*Figure 7E*). Together, these results indicate that CCS movement, CME and Actin dynamics are altered specifically in the presence of pathogenic TDP-43 and is phenotypically similar to our observations with HTT Q138.

## TDP-43 aggregates result in the alteration of cellular physical properties

To elucidate the effect of cell's pathogenic aggregates on its mechanical response, we performed AFM experiments on single cells harboring different types of aggregates- Aβ–42–2 x, FUS R521C, αSynA30P, αSynA53T or TDP-43, similar to what was described in *Figure 5*. Our PIV analysis has shown that, with the exception of HTT Q138 and TDP-43, the actin dynamics does not get altered in the presence of other aggregates (*Figure 7*). Both CME and filopodia formation were significantly altered in the presence of HTT Q138 and TDP-43, while Aβ–42–2 x, FUS R521C, αSynA30P and αSynA53T had no significant effects on CME or filopodia formation compared to wild-type cells (*Figures 6 and 7*). The AFM experiments showed a direct correlation of the cell's physical properties with the state of actin organization and CME. Young's modulus for cells having pathogenic aggregates of Aβ–42–2 x,

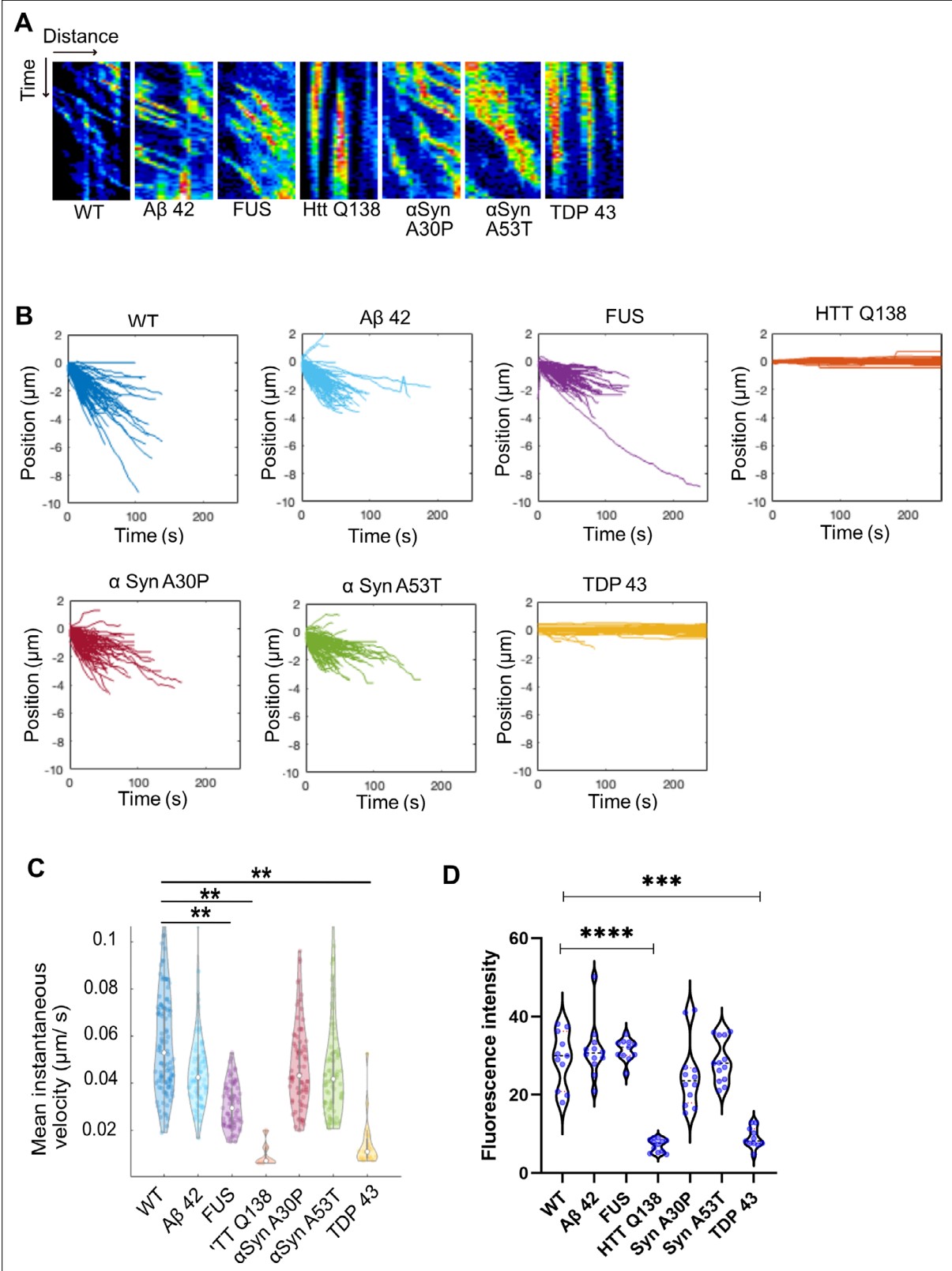

**Figure 6.** Cells expressing pathogenic TDP-43 show deficits in CCS movement. (**A**) Kymographs showing the movement of CCSs from wild type hemocytes, or in the presence of the indicated proteins. (**B**) Position vs. time plots of CCSs in WT hemocytes and hemocytes expressing aggregating proteins. The positions of CCSs with respect to the cell centre are plotted. Negative positions indicate movement towards the cell centre. The total number of vesicles used for quantitation from ten cells are as follows: WT – 117; Aβ42–2 x – 105; FUSR521C – 83; HTT Q138 – 103; αSynA30P – 91;

*Figure 6 continued on next page*

*Figure 6 continued*

αSynA53T - 95 and TDP 43–106. (**C**) Violin plots of mean instantaneous velocities of CCSs in WT hemocytes and hemocytes expressing aggregating proteins. Asterisks represent a significant difference from WT (p<0.05, Kruskal Wallis test for non-parametric data). (**D**) Quantification of fluorescence intensity of internalized mBSA in hemocytes expressing the indicated aggregating proteins n=10. Statistical significance was determined using Kruskal-Wallis test followed by post hoc Dunn's multiple comparison test. For WT vs HTT Q138 p-value <0.0001, **** and for WT vs TDP 43 p value = 0.0005 ***.

The online version of this article includes the following video and figure supplement(s) for figure 6:

**Figure supplement 1.** Graph showing fluorescence intensity of CLC GFP upon FRAP of individual CCSs.

**Figure 6—video 1.** Video showing time lapse imaging of a hemocyte containing Aβ42 aggregates expressing Clathrin light chain tagged with GFP over 5 min at 5 s intervals.

https://elifesciences.org/articles/98363/figures#fig6video1

**Figure 6—video 2.** Video showing time lapse imaging of a hemocyte containing FUS R521C aggregates expressing Clathrin light chain tagged with GFP over 5 min at 5 s intervals.

https://elifesciences.org/articles/98363/figures#fig6video2

**Figure 6—video 3.** Video showing time lapse imaging of a hemocyte containing α-SynA30P aggregates expressing Clathrin light chain tagged with GFP over 5 min at 5 s intervals.

https://elifesciences.org/articles/98363/figures#fig6video3

**Figure 6—video 4.** Video showing time lapse imaging of a hemocyte containing α-SynA53T aggregates expressing Clathrin light chain tagged with GFP over 5 min at 5 s intervals.

https://elifesciences.org/articles/98363/figures#fig6video4

**Figure 6—video 5.** Video showing time lapse imaging of a hemocyte containing TDP-43 aggregates expressing Clathrin light chain tagged with GFP over 5 min at 5 s intervals.

https://elifesciences.org/articles/98363/figures#fig6video5

FUS R521C, α-SynA30P and α-SynA53T remained similar to WT cells (*Figure 8A*). The stiffness was altered in case of TDP-43, similar to what was seen in HTT Q138, showing a direct correlation of cell's modulus of elasticity with actin dynamics and CME transport.

In our study, we demonstrate that specific pathogenic aggregates alter the physical properties of cells, by directly impacting actin dynamics and hence the CME process (*Figure 8B*). The PIV analysis suggests that actin dynamics in the presence of HTT Q138 and TDP-43 were restricted compared to WT cells, leading to compromised CCS movement (*Figure 1* and *Figure 6*). The cells with HTT Q138 and TDP-43 aggregates were observed to have increased Young's modulus compared to WT cells indicating stiffening of cells. This is also observed to be the result of altered actin dynamics. Recovery of stiffness of the cells upon overexpression of actin binding proteins (Hip1 or Arp3) reverts cells towards the WT, providing strong support to our hypothesis. More importantly, our AFM results show that mechanical response of cells can be used as a biomarker for actin mediated altered cell behavior in the presence of pathogenic protein aggregates. Further, our results demonstrate that the toxicity caused by pathogenic protein aggregates may be due to a reorganization of other proteins, organelles and cellular pathways rendering them dysfunctional (*Figure 8B*). Rescuing some of these functions may indeed work towards reducing neurotoxicity even in the continued presence of pathogenic aggregates.

## Discussion

Endocytosis has been shown to be affected in the context of neurodegeneration caused by pathogenic aggregates. However, it is as yet unclear as to how exactly such aggregates alter the process of endocytosis. It is well-documented that pathogenic aggregates affect the function of both neuronal and non-neuronal cells. However, whether similar processes are affected across multiple cell types remains poorly studied. Our work assesses CME by tracking clathrin light chain-positive structures in live cells in the presence of aggregating proteins responsible for neurodegeneration. The extent of observed change in CCS speed and directionality varied depending on the mutant protein present, suggesting that specific interactions may be responsible for the phenotype, rather than merely the physical presence of an aggregate in the cell. The most drastic changes resulting in a complete block in CCS movement were observed in the context of HTT Q138 and TDP-43. While the speed of CCS movement was also marginally altered in the presence of other mutant proteins, a complete block was

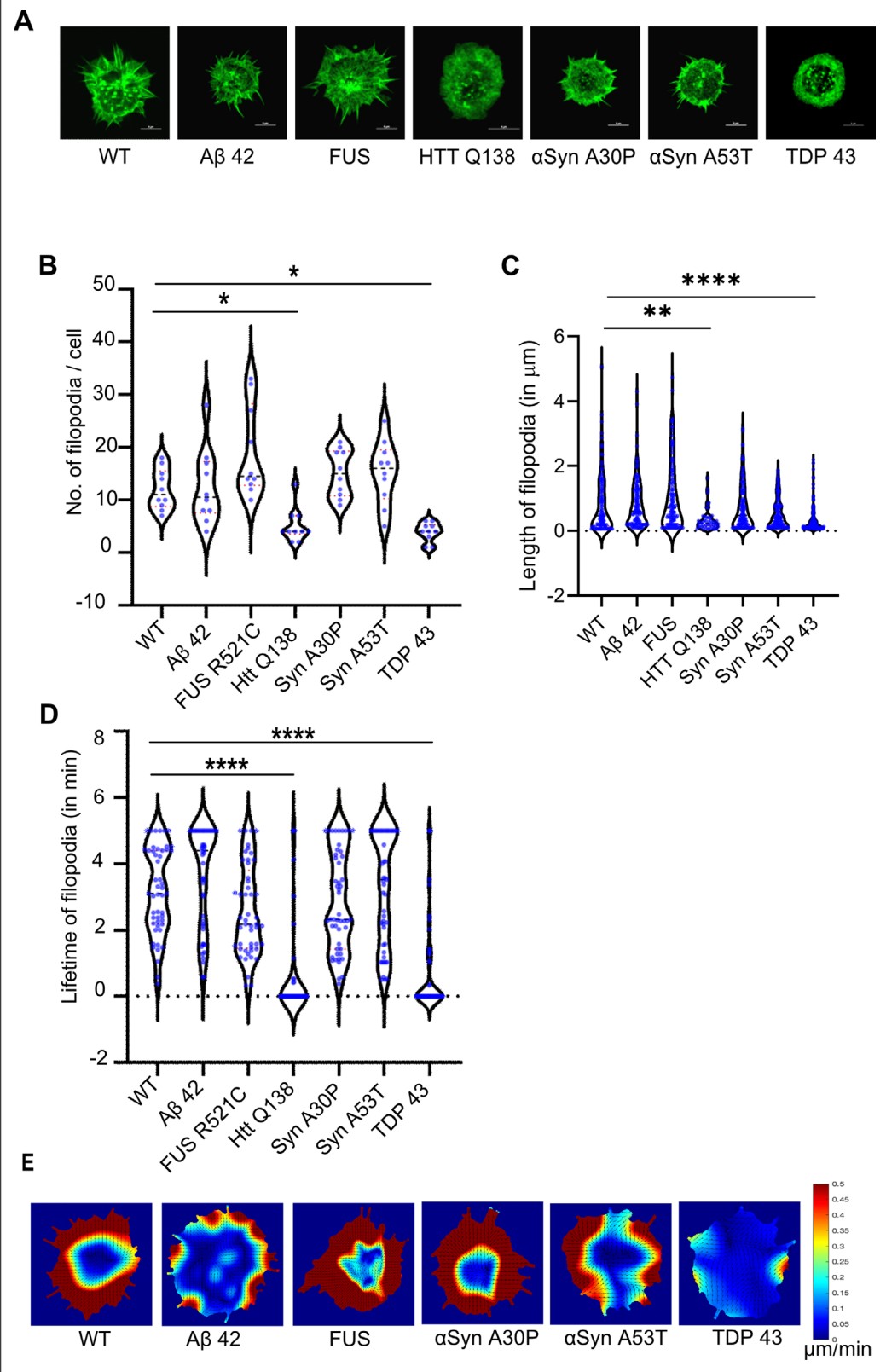

**Figure 7.** Cells expressing pathogenic TDP-43 show altered Actin dynamics. (**A**) Representative micrographs showing the organization of actin marked by Lifeact-GFP in hemocytes. Scale bar 5 μm. (**B**) Graph showing the quantification of the number of filopodia per cell (n=10 cells for each condition). Statistical significance was determined using Kruskal-Wallis test followed by post hoc Dunn's multiple comparison test. For WT vs HTT

*Figure 7 continued on next page*

*Figure 7 continued*

Q138 p-value = 0.0448,* and for WT vs TDP43 p-value = 0.0135, *. (**C**) Graph shows the quantification of length of filopodia in micrometres (n=10 cells for each condition). Statistical significance was determined using Kruskal-Wallis test followed by post hoc Dunn's multiple comparison test. For WT vs HTT Q138 p-value = 0.0043, ** and for WT vs TDP 43 p value <0.0001, ****. The number and length of filopodia were quantified using the Filoquant plugin in ImageJ. (**D**) Violin plots of the lifetime of filopodia in cells expressing Aβ42–2 x, FUSR521C, αSynA30P, αSynA53T, HTT Q138, TDP-43 compared to WT cells. Kruskal-Wallis test followed by post hoc Dunn's multiple comparison was performed to calculate the p value. (p-value <0.0001). (**E**) PIV analysis performed on LifeAct-GFP-expressing cells in the presence of the indicated pathogenic aggregating proteins to highlight the direction and magnitude of actin flow.

The online version of this article includes the following video and figure supplement(s) for figure 7:

**Figure supplement 1.** Micrograph showing representative images obtained from Filoquant for quantification of number and length of filopodia in presence of the indicated mutant protein.

**Figure 7—video 1.** Video showing time lapse imaging of a hemocyte containing Aβ42 aggregates expressing Lifeact tagged with GFP over 5 min at 5 second intervals.
https://elifesciences.org/articles/98363/figures#fig7video1

**Figure 7—video 2.** Video showing time lapse imaging of a hemocyte containing FUS R521C aggregates expressing Lifeact tagged with GFP over 5 min at 5 s intervals.
https://elifesciences.org/articles/98363/figures#fig7video2

**Figure 7—video 3.** Video showing time lapse imaging of a hemocyte containing α-SynA30P aggregates expressing Lifeact tagged with GFP over 5 min at 5 s intervals.
https://elifesciences.org/articles/98363/figures#fig7video3

**Figure 7—video 4.** Video showing time lapse imaging of a hemocyte containing α-SynA53T aggregates expressing Lifeact tagged with GFP over 5 min at 5 s intervals.
https://elifesciences.org/articles/98363/figures#fig7video4

**Figure 7—video 5.** Video showing time lapse imaging of a hemocyte containing TDP 43 aggregates expressing Lifeact tagged with GFP over 5 min at 5 s intervals.
https://elifesciences.org/articles/98363/figures#fig7video5

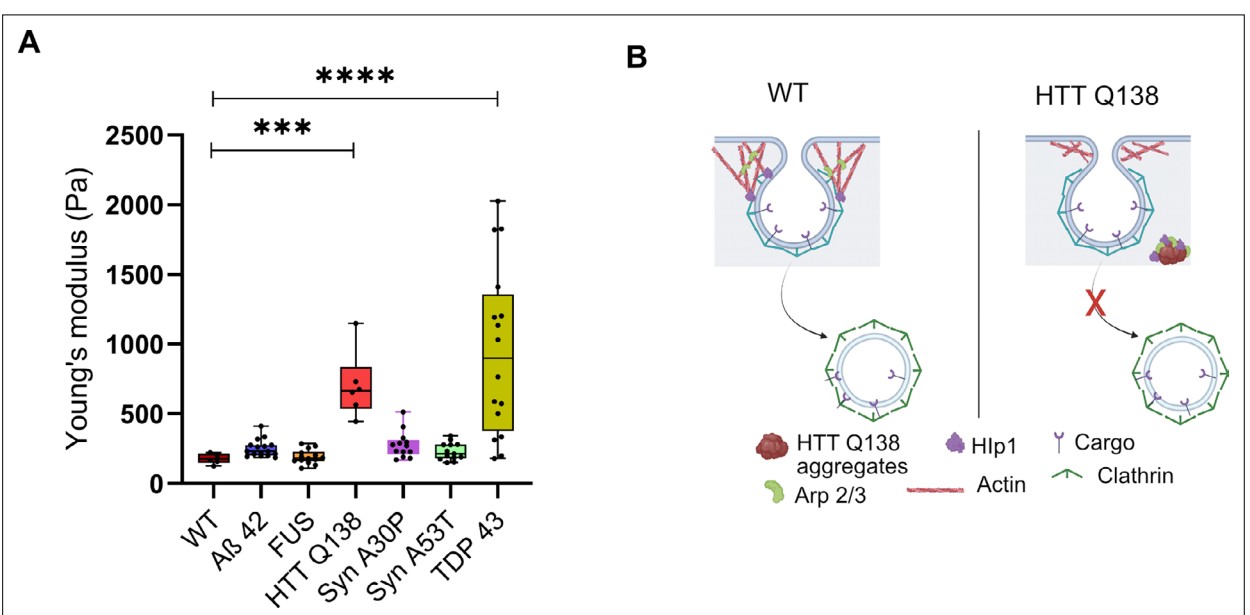

**Figure 8.** TDP-43 aggregates result in the alteration of cellular physical properties. (**A**) Box plot showing the Young's modulus of all the indicated cell types. AFM was performed on 10 cells of each cell type ***, p-value = 0.0005 and ****, p-value <0.0001. Statistical significance was determined using Kruskal-Wallis test followed by *post-hoc* Dunn's multiple comparisons. (**B**) Model showing alteration in actin organization and clathrin-mediated endocytosis in the presence of Huntingtin aggregates due to sequestration of Hip1 and components of the Arp2/3 complex.

only observed in the presence of HTT Q138 and TDP-43 (*Figure 1* and *Figure 6*). Furthermore, these alterations in CCS movement also correlated with changes in actin dynamics as reflected by changes in filopodia length, lifetimes and actin flow (*Figure 2* and *Figure 7*). Inhibition of polymerization of the actin cytoskeleton either by chemicals (LatA/ CytoD) or by genetic means (profilin knockdown) also led to stalled CCS movement, similar to what was observed in the presence of HTT Q138 or TDP-43 aggregates. Our results also revealed that components of the Arp2/3 complex were sequestered by HTT Q138 aggregates (*Figure 3*), and that overexpression of Arp2/3 complex components rescued CCSs movement and filopodia formation even in presence of HTT Q138 aggregates (*Figures 4 and 8B*). Conversely, inhibiting the function of Arp2/3 either by CK666 or by RNAi resulted in stalled CCS movement (*Figure 3*). Reduced barbed end labeling of actin was also observed in the presence of HTT Q138 (*Figure 2*). Together, these indicate that the branching function of Arp2/3 is essential for CME, and that the altered function of Arp2/3 by HTT Q138 aggregates may be one of several possible mechanisms by which CME is inhibited in Huntington's disease. In support of this, both CCS movement in the presence of HTT Q138 aggregates and neurodegeneration could be rescued by overexpression of Hip1, Arp3 or the chaperone, Mrj (*Figure 4*). This is in agreement with previously published data which demonstrates that overexpression of specific proteins that are sequestered into Huntingtin aggregates may help to reduce the toxic effects of the aggregates (*Hosp et al., 2017*). Our study suggests that overexpressing Hip1 and Arp2/3 ameliorates Huntingtin aggregate-induced loss of function and restore cellular processes (*Figure 8B*).

Previous reports have demonstrated a tight correlation between the availability/distribution of Hip1R, Arp2/3 complex and endocytic rate and efficiency (*Akamatsu et al., 2020*). As the available number of Arp2/3 molecules increase, internalization depth increased in a shorter span of time. Further, while the distribution of Arp2/3 did not affect endocytic efficiency, the distribution of Hip1R regulated the endocytosis rate and extent, presumably by regulating the recruitment and distribution of Arp2/3 complexes on the surface of the growing endocytic pit. In vitro experiments have also revealed that the ratio of branching to capping protein regulates stiffness of the actin network. Network stiffness increased as either capping or branching protein concentration was increased (*Pujol et al., 2012*). The authors observed that this was in contrast to what was predicted, as increased capping should result in shorter filaments which should result in a softer network. However, this points to the fact that network stiffness is dependent on regulating a balance between branching and capping, and that an alteration in these may affect stiffness and endocytic efficiency.

Previous studies looking at the FRAP of the clathrin light chain reveal that considerable exchange exists between the membrane associated and soluble pools of the protein (*Wu et al., 2001*; *Wu et al., 2003*). FRAP of the clathrin light chain in the presence of HTT Q138 and TDP-43 aggregates showed an incomplete recovery at CCSs (*Figure 3—figure supplement 1*, *Figure 6—figure supplement 1*). Furthermore alteration of actin cytoskeleton also shows delayed recovery of clathrin (*Figure 3—figure supplement 1*). These results indicate that availability/exchange of clathrin at the sites of CME is compromised in both these cases, in a manner hitherto unknown.

Our results suggest that the changes observed in CME in the context of HTT Q138 and TDP-43 may be dependent on the actin cytoskeleton, whereas this may not be the case in the context of other aggregates. Finally, we also determine that the mechanical response as measured by cellular stiffness is also significantly altered in the presence of HTT Q138 and TDP-43 (*Figures 5 and 8*). Previous work from our lab has demonstrated that a loss of CME in embryonic stem cells results in an altered organization of the actin cytoskeleton and an increased cellular stiffness (*Mote et al., 2020*). The mechanical response by measuring the Young's modulus, observed in the presence of HTT Q138 and TDP-43 aggregates also appears to be linked to the altered and less dynamic state of actin organization. The branched actin network is known to generate force required in many cellular processes including endocytosis, with alterations in this network directly limiting force transmission (*Goode et al., 2015*; *Lacy et al., 2018*; *Planade et al., 2019*). Our results show that regulating the cells' physical properties, which is principally due to the actin network stiffness, by overexpressing Hip1 or Mrj, or by transient treatment with LatA could restore the stiffness of cells to WT levels even in presence of HTT Q138 aggregates. This suggests that a 'Goldilocks' state of the branched actin network is required for sufficient and efficient force generation during CME, where it is neither too soft, nor too stiff, but in an intermediate, dynamic state. The rescue of neurodegeneration driven by pathogenic Huntingtin by the overexpression of actin binding proteins, indicates that the remodelling of the actin cytoskeleton contributes

largely to the aggregate-driven neurotoxicity. Further, the similarity in phenotypes observed in the context of CCS movement across different cell types from multiple organisms throws up interesting possibilities for screening potential molecular candidates in the context of Huntington's disease.

Why do various aggregating pathogenic proteins display such a variety of phenotypes with respect to CME and actin organisation? The presence of such altered and varied mechanisms have been reported even in the context of a single pathogenic protein. Overexpression of the monomeric versus the dimeric form of α-synuclein in the lamprey giant reticulospinal synapses demonstrated a block of CME at different stages indicative of their ability to interfere at different stages of CME, suggestive of a differential interaction with proteins (*Medeiros et al., 2017*). TDP-43 knockdown has also been shown to alter the number and motility of Rab11-positive recycling endosomes (*Schwenk et al., 2016*), while the loss of HTT has been shown to result in the altered activation status of cofilin, thereby resulting in altered actin dynamics (*Wennagel et al., 2022*). Further, the toxicity associated with TDP-43 was enhanced upon inhibiting endocytosis (*Liu et al., 2017*).

Recent work has highlighted a role for Nwk/FCHSD2, Dap160/Intersectin and WASP in regulating actin assembly at synapses (*Del Signore et al., 2021*). Interestingly, Intersectin has been shown to associate with mutant forms of HTT and enhance its aggregation resulting in increased neuro-toxicity (*Scappini et al., 2007*). It remains unknown whether TDP-43 also exhibits similar interactions, thus resulting in similar phenotypes with respect to CCS dynamics and cellular stiffness. Additionally, it is also unknown whether other mutant proteins involved in aggregation and neurodegeneration interact with specific proteins involved in actin organization and endocytosis at the synapse, which in turn may affect their aggregation. These would form the basis for future studies and may also shed light on whether the endocytic phenotypes in these aggregate-containing cells are due to similar or different molecular mechanisms.

## Materials and methods

### *Drosophila* strains and crosses

*Drosophila* strains were grown under controlled conditions in incubator at 25°C maintaining day and night cycle.

Recombinant fly lines were generated by crossing CgGAL4 flies (RRID:BDSC_27396) with Clathrin light chain (Clc) tagged with GFP (UAS-Clc-GFP) (RRID: BDSC_7107) or with Lifeact tagged with GFP (UAS-Lifeact-GFP) (RRID: BDSC_56842) flies. GFP-positive larvae were selected and flies emerging from these larvae were balanced with CyO and maintained as recombinant lines for further use. These were crossed with aggregate protein-expressing flies (UAS-mRFP-Htt *Weiss et al., 2012*, UAS-TDP-43 *Lanson et al., 2011*, UAS-FUSR521C *Lanson et al., 2011* UAS-2xAβ–42 *Casas-Tinto et al., 2011*, UAS-αSynA30P *Auluck et al., 2002* (RRID:BDSC_8147), UAS-αSynA53T *Auluck et al., 2002* (RRID:BDSC_8148)) and hemocytes from third instar larvae were collected.

### Hemocyte isolation

Hemocytes used for all experiments were isolated from third instar larvae. Single larvae were dissected with the help of fine tweezers, in Schneider's medium (S2 cell medium) (Catalog no. 21720024, Gibco) and hemocytes were collected in a 35 mm glass bottom dish.

### mBSA internalization assay

Maleylation of Bovine serum Albumin (BSA, MP biomedicals) was performed as described earlier (*Guha et al., 2003*). Fluorescent tagging of mBSA was performed using Alexa Fluor 594 Microscale Protein Labelling kit (Catalog no. A30008, Invitrogen, Molecular Probes) as per manufacturer's protocol. For mBSA uptake, larvae were dissected and hemocytes were collected in serum free Schneider's *Drosophila* medium. Cells were incubated for 10–15 min at room temperature to allow them to settle down. Cells were then incubated with mBSA at a concentration of 1 µg/ml for 35 min and washed several times with ice cold PBS. Fixation of hemocytes was done using 4% PFA for 20 min and cells were washed before imaging.

### Mammalian cell culture and transfection

HEK293T cells (RRID: CVCL_0063) were maintained in DMEM media (Gibco) supplemented with 10% FBS. SH-SY5Y cells (RRID: CVCL_0019) were maintained in DMEM/F12 (1:1) (Gibco) media

supplemented with 10%FBS. Cells were plated in glass bottom dishes and transfected with CHC-GFP plasmid (RRID:Addgene_59799) along with HTT Q138 mRFP cloned in pcDNA 3.1 using Lipofectamine 2000 (invitrogen). Cells were imaged 48 hr after transfection.

## Actin monomer labeling

Actin monomers were isolated from rabbit muscle acetone powder using the standard procedure (*Pardee and Spudich, 1982*; *Pollard, 1984*). Subsequently, these monomers were tagged with Cy5 Alexa-488 maleimide (Thermo Fisher Scientific) in excess (3–5 M) within a labelling buffer (composed of 5 mM Tris–HCl, pH 8, 200 µM ATP, and 100 µM $CaCl_2$) on ice for 10 min, following the protocol established by *Hansen et al., 2013*. The labelling reaction was halted by the addition of 10 mM DTT. Afterward, the monomeric actin was centrifuged at 350,000 × *g* (using a TLA 120.2 rotor from Beckman Coulter) for 30 min to eliminate aggregated proteins and insoluble fluorescent dye. The resulting supernatant was polymerized at room temperature (25 °C) in the presence of 10 X KMEI buffer (composed of 50 mM KCl, 1 mM $MgCl_2$, 1 mM EGTA, 10 mM imidazole, pH 7.0, and 1 mM ATP). The polymerized actin was then subjected to ultracentrifugation at 350,000 × *g* for 1 hr. Following polymerization, the actin filaments were fragmented in the presence of 2 mM Tris–HCl, pH 8.0, 0.2 mM ATP, and 0.1 mM $CaCl_2$, and allowed to depolymerize for 3–4 days at 4 °C. After depolymerization, the labeled actin was again ultracentrifuged to eliminate actin seeds and aggregates. To remove any remaining unconjugated dye, the depolymerized actin underwent buffer exchange using a PD10 desalting column.

## Actin barbed end labeling and quantification

Actin labeling was done using the previously described protocol (*Marsick and Letourneau, 2011*). In brief, cells grown on glass-bottomed dishes were permeabilized using a permeabilization buffer containing 10 mM PIPES, 138 mM KCL, 4 mM MgCl2 and 1% BSA (pH = 6.9) for 1 min. Saponin was added at a final concentration of 0.025% just before use. Permeabilization buffer was removed gently and replaced with permeabilization buffer containing 450 nM Cy5 Alexa-488 conjugated actin and 0.1 mM ATP for 5 min, followed by fixation in 4% paraformaldehyde for 5 min. Incorporation of Cy5 Alexa-488 actin onto barbed ends was visualized at 2.6 X zoom using 100 X plan Apo oil immersion objective with 1.49 NA (Nikon). Quantification of intensity of labeled barbed ends was done using ImageJ.

## Cortical actin analysis

To quantify thickness and actin intensity at the cell cortex expressing Lifeact GFP in the background of Htt Q15/Q138, a single line was drawn across the cell and thickness and intensity was obtained using NIS Nikon analysis software. Data was plotted using GraphPad Prism.

## Rescue of photoreceptor neurodegeneration in *Drosophila* eye

Long glass multiple reporter (LGMR) - GAL4 flies were crossed with UAS -HTT Q127 expressing fly line. Virgin flies were then crossed with the balancer line CyO/ Sco to make a transgenic line. LGMR-GAL4 >UAS HTT Q127/Cyo virgin flies were then crossed with UAS CD8 GFP (Positive control), UAS-hHip1 (RRID: BDSC_66279), UAS-Arp3-GFP (RRID: BDSC_39722) or UAS-Mrj. Photoreceptor rescue was determined in adult flies.

## Inhibitor treatment

Hemocytes were treated with the following inhibitors at the mentioned concentrations: Latrunculin A (Sigma Aldrich cat. no. L5163) and Cytochalasin D (Sigma Aldrich, cat no. C8273), were used at a final concentration of 1 µM. Latrunculin A was added at the time of imaging, while cells were treated for 1 hr with Cytochalasin D prior to imaging. CK666 (SigmaAldrich, cat. no. SML0006), was used at a final concentration of 50 µM, and cells were treated for 40 min prior to imaging. SMIFH2 (Sigma-Aldrich, cat. No. 344092), was used at a final concentration of 50 µM and cells were incubated for 15 min at room temperature before time lapse imaging.

## Imaging and image analysis

Time lapse imaging of hemocytes were performed along a single X-Y plane close to the glass cover-slip to look at clathrin light chain and actin dynamics. Imaging was carried out for 5 min at 5 s intervals using a Nikon Ti Eclipse confocal microscope. HEK293T and SHSY5Y expressing CHC GFP were imaged for 3 min at 5 min intervals and kymographs were generated using imageJ. All images were acquired at 2.6 X zoom using 100 X plan Apo objective with 1.49 NA, using manufacturer's software. To perform FRAP, hemocytes were isolated in S2 medium in glass-bottomed confocal dishes. ROI was drawn on a single non-motile CCS at the periphery of cells and photo-bleached using 10% laser power of the 488 nm laser with 100 X, 1.49 NA oil immersion objective and pinhole of 1.2 AU. Pre-bleach time lapse images were acquired for 15 frames at 5.66 s interval followed by bleaching for one frame for 0.971 s duration. To look at the recovery of Clc-GFP, CCSs were then imaged for 50 frames at 5.66 s interval. FRAP analysis was done using the online easyFRAP tool *Koulouras et al., 2018* and mean intensity from five independent experiments was plotted.

Kymographs were generated using Image J software. CCSs were tracked using the Low Light Tracking Tool plugin in Fiji/ImageJ (*Krull et al., 2014*) (RRID: SCR_002285). The resulting tracks were analysed and quantified using custom functions written in MATLAB, Mathworks. The mean instantaneous velocities of CCSs were quantified by calculating the distance moved by an individual CCS for each 5 s interval of the time-lapse images, and averaging these instantaneous velocities for that CCS. All plots were generated in MATLAB. For tests of significance, data were first checked for normality using the chi2gof function and MATLAB.

Length and number of filopodia were quantified using Filoquant software, keeping the threshold constant across all the cells. Lifetimes of filopodia were calculated by manually calculating the duration that a filopodia persisted in a cell.

PIV analysis for actin flow was performed as previously described (*Yolland et al., 2019*) using the following PIV code: https://github.com/stemarcotti/PIV (*Marcotti, 2022*). Source size was set to 0.5 µm, while search size was set to 1.0 µm. All other parameters were kept the same as in the given code. PIV analysis was performed on atleast 5 cells in each case.

## Quantification of 2D flow-fields from the time-lapse movies

To understand the movement of CCSs, we computed the 2D velocity flow-field $\vec{v}(x, y)$ by performing a Particle Image Velocimetry (PIV) (*Raffel et al., 2007*) analysis on the time-lapse movies obtained from the experiments. Prior to the PIV analysis, the movies were corrected for drift errors using StackReg, an ImageJ plugin, after which an intensity correction (see Extended methods for details) was applied to account for the bright background clathrin intensity which could significantly modify the nature of the flow-field.

## PIV analysis of the corrected time-lapse movies

The PIV analysis was performed using the PIV lab toolbox via Matlab (https://pivlab.blogspot.com/). Before running the PIV analysis, the region-of-interest was chosen around the cell boundary and a common mask was applied to each frame to ensure that there was no spurious intensity from regions outside the cell boundary. To calculate the cross-correlation during the PIV analysis, we chose the Fast Fourier Transform (FFT) window deformation, with three passes starting from 512 pixels and with the final interrogation area of 64 pixels. After performing the PIV analysis over each movie, we performed a post-processing over each time-frame. First we applied a standard deviation filter with threshold value of 4 followed by a local median filter with threshold value 4. This ensured that we excluded velocity vectors with large variabilities. We also apply the interpolation filter which returns interpolated vector values for the missing data. Finally, we also applied a one-step spatial smoothing of the velocity flow-field for each frame. Note that we have checked our results by performing the PIV analysis (i) without applying the interpolation filter at each frame, and, (ii) with a finer binning where the final interrogation area taken to be 32 pixels. For both these cases, we obtain a similar radial flow profile $V_r(r)$ (discussed in the following sections). We note from the radial speed profiles that for the Wild Type cells, the maximum radial speed, which is attained near the cell membrane is $V_r(r \to R) \approx 0.005 \mu m/sec$. This value is lower than the speed values estimated from the particle tracking experiments (*Figure 6C*). This is not surprising as the PIV analysis gives an estimate of the hydrodynamic flow-fields by computing the average displacement of a large number of particles. On

the hand other individual particle tracking is more precise and was done over selective trajectories taken close to the membrane and also across a large number of cells. Thus it is expected that PIV analysis will underestimate the speed magnitudes compared to the original value and the degree of which depends on the presence of spurious noise in the movies. However on the other hand, PIV analysis efficiently captures the large-scale continuum flow-field and may reveal more about the underlying physical principles that lead to such collective flows.

## Time-averaged steady-state flow profile

To further quantify the flow profiles from the PIV analyzed movies, we made a steady-state assumption, since the overall flow pattern did not change over the duration of movies.

Under the steady-state assumption, we obtain the time-averaged spatial flow-field profile by averaging over the entire time duration $T = N\Delta t$ of the movie:

$$\vec{V}(i,j) = <\vec{v}(i,j,k)>_k = \frac{1}{N}\sum_{l=1}^{N}\vec{v}(i,j,l).$$

Here $\vec{v}(i,j,k)$, is the flow-vector, where each $(i,j)$ value indicate the pixel coordinates, and the index $k$ indicates the time-frame. Here $N$ is the number of time-frames and $\Delta t$ is the inverse frame-rate. Finally using the pixel to micron conversion information (*Supplementary file 1*), we obtain the average spatial flow-field $\vec{V}(x,y)$ where $(x,y) = (i\Delta x, j\Delta y)$. In *Figure 1—figure supplement 1*, we show the time-averaged flow-field obtained from PIV analysis plotted on top of the corrected intensity field for the wild type cell.

## Quantification of PIV data

From the time-averaged spatial flow-field profile for the Wild type (*Figure 1—figure supplement 1*), it is evident that the CCSs flow radially inwards from the cell membrane to the cell center while in the mutants such as HTT Q138 and cells treated with LAT-A, this movement is severely affected. To quantify the nature of this radial movement and to compare between the different experimental cases, we obtained the radial speed profile from the time-average spatial flow field $\vec{V}(x,y)$. As a first step, we identified the cell centre $(x_c, y_c)$. The flow-field was then translated to new coordinates $x \rightarrow (x - x_c), y \rightarrow (y - y_c)$. We then performed a transformation to polar coordinates $(r, \theta)$, where $r = \sqrt{x^2 + y^2}$, and $\theta = tan^{-1}(y/x)$ and computed the radial component of the speed, which is given as $V_r(r, \theta) = \vec{V}(x,y) \cdot \hat{r} = (V_x cos\theta + V_y sin\theta)$. Note that one can also similarly compute the polar component $V_\theta(r, \theta)$ which is negligible for the Wild type, and therefore we do not quantify it. Next we obtained the radial flow profile purely as a function of the radial distance $r$, namely $V_r(r) = <\frac{1}{2\pi}\int_{-\pi}^{\pi}d\theta V_r(r, \theta)>_{r+\Delta r}$. Note that the $V_r(r)$ value was obtained by averaging over all possible $\theta$ values over an annulus between $[r, r + \Delta r]$. We choose $\Delta r = 60$ pixels such that $0.5\mu m \leq \Delta r \leq 1.2\mu m$ (see *Supplementary file 1*). Note that to obtain $V_r(r)$, we assumed that the flow-field has a polar symmetry. While the cells are not exactly circular and a more rigorous quantification in that case would be to estimate the exact cell boundary contour and perform the polar-angle averaging over concentric contours. However, the nature of the flow-field will remain unaffected and we therefore proceeded with this simplified definition.

Finally we quantified the directionality of the flow-field by computing the distribution of the velocity direction relative to the polar direction. To obtain this distribution, we first constructed the variable $\theta_{dir} = (\theta_v - \theta)$, where $\theta_v = tan^{-1}(V_y/V_x)$. For a pure radial flow, we will have either $\theta_{dir} = 0$ or $\theta_{dir} = \pi$, depending on whether the flow is radially inward or outward. For purely circular motion, $\theta_{dir} = \pm\pi/2$, depending on whether it is clockwise or counter-clockwise. Once $\theta_{dir}$ is obtained for each pixel $(r, \theta)$, we compute the corresponding normalized histogram $P(\theta_{dir})$.

## S2 cell culture and transfection

Approximately $1\times10^6$ cells were plated in 12 well plates in S2 cell medium (Schneider's medium). Cells were transfected with Huntingtin mRFPQ15, Huntingtin mRFPQ138 and Actin-GFP plasmid along with PAC-GAL4 vector at a concentration of 200 ng each, using PEI (Polyethyleneimine). S2 cells were grown at 25°C in a humidified incubator in 5% $CO_2$.

## FRAP and analysis

To perform FRAP, hemocytes were isolated in S2 medium in glass bottom plates. ROI was drawn on single non-motile CCS at the periphery of cells and photo-bleached using 10% laser power of 488 nm laser with 100 X, 1.49 NA oil immersion objective and pinhole of 1.2 airy unit. Pre-bleached time lapse images were acquired for 15 frames at 5.66 s interval followed by bleaching for one frame for 0.971 s duration. To look at the recovery of Clc-GFP, CCSs were imaged for 50 frames at 5.66 s interval. FRAP analysis was done using the online easyFRAP tool *Koulouras et al., 2018* and mean intensity from five independent experiments was plotted using Graphpad Prism.

## Atomic force microscopy and nanoindentation experiments

The nanoindentation experiments on single cells were performed using JPK Nanowizard II AFM from Berlin, Germany. The tipless cantilevers from MikroMasch, Bulgaria (model no.- HQ:CSC38/tipless/Ce-Au) were used. The tipless commercial cantilever was customized by attaching a 5 μm (diameter) glass-bead using lift-off method as described previously *Singh et al., 2024*. Briefly, the resin and hardener of a two-component epoxy (Araldite Klear, India) were mixed in 1:0.8 ratio and stirred until it turn a milky color. A few tiny glue-droplets were decorated on a clean glass slide. The cantilever was approached on one of the droplets carefully such that the cantilever's free end should just touch the droplet's topmost layer and pick a tiny amount of glue, while avoiding dipping the cantilever. The cantilever was retracted by ~20 μm. It was brought near the area where the glass-beads were sparsely coated on the same glass slide. The cantilever was brought into contact on a clean area a couple of times at different locations on the glass slide to reduce the excess amount of glue. The cantilever was brought onto a clean glass bead and aligned in such a way that the bead would attach at the desired position. Once aligned, it is approached using the auto approach mechanism with the setpoint of 0.15–0.2 Volts. Once the auto approach was done, the setpoint was gradually increased to 1.0 Volt. The glue is allowed to cure for 20–30 min. The cantilever was retracted and it was confirmed that the bead remained attached. The position of bead attachment was determined by Scanning Electron Microscope (SEM) imaging. We only used the cantilevers with beads attached in the middle along the cantilever width.

The photodetector sensitivity and cantilever force constant were determined before every experiment. The detection sensitivity (in nm/V) was determined by obtaining a slope of approach curve at the cantilever-glass slide deep contact region. The force constant was determined using thermal tuning method *Heim et al., 2004* available in the software. The cantilevers with a typical force constant of 0.2 N/m were used for experiments.

### Nanoindentation experiment

Hemocytes were isolated from third instar larvae and plated onto Concanavalin-coated 22 mm circular glass coverslips, which were freshly cleaned and wiped with methanol to avoid unwanted dirt on the surface before experiments. Cells were incubated for 30 min at room temperature to allow proper adherence to the coverslip. The approach-retract experiments were performed on a single cell with ~2 μm/s speed on a 1x1 μm2 area using a 6x6 grid with a sampling rate of 2 kHz. We allowed a rest period of 2 s between two successive force curves for the cell to recover its original shape. The experimentally recorded parameters (raw data) were cantilever deflection (dcant, in Volts) and the base piezo displacement (dpz, in Volts). The piezo displacement corresponds to movement of the cantilever base. dcant and dpz were converted to the unit of meters by multiplying them with photodetector sensitivity and base-piezo sensitivity, respectively. The force (F) on the cantilever/sample was calculated by multiplication of dcant with the cantilever force constant. The cell deformation was determined by subtracting cantilever deflection from the base-piezo displacement.

### Data analysis

The force curves were fitted with Hertz model by fitting Sneddon's formula. The relation between force ($F$) and deformation ($\delta$) of a linear elastic material is as follows:

$$F = \frac{E_{Hertz}}{1 - \nu^2} \left[ \frac{a^2 + R^2}{2} \ln \frac{R + a}{R - a} - aR \right]$$

and

$$\delta = \frac{a}{2} ln \frac{R+a}{R-a}$$

Where $E_{Hertz}$ is the Young's modulus. $R$, $a$ and $\nu$ are the radius of the indenter, radius of the contact circle and sample's Poisson's ratio of respectively.

## Extended methods

### PIV - drift correction

Often during the process of image acquisition, there are external mechanical perturbations which lead to finite drift in the centre of mass position of the cell, with respect to the image frame, over the duration of imaging. It is therefore absolutely necessary to ensure that the internal flow vectors are corrected for drift in the cell because of the global movement. We rectified the global offset of the cell centre of mass in the time-lapse movies using StackReg, an ImageJ plugin, which can effectively correct any centre of mass movement. More details about the plugin can be found at https://bigwww. epfl.ch/thevenaz/stackreg/.

Once all frames from the time-lapse movies were drift corrected, we performed the next step, namely intensity correction, which is discussed below.

### Intensity correction

One of the main challenges we faced while analyzing the time-lapse movies is the presence of static clathrin intensity in the background, the source of which could be clathrin patches or newly translated clathrin protein. PIV analysis of the movies in presence of this fluorescence leads to incorrect estimation of flow field. This is because the stationary fluorescent particles are considerable in number and size and show up in autocorrelation thereby significantly underestimating the flow-fields. To reduce the effect of stationary intensity during the PIV analysis, we attempted to remove the effect of stationary fluorescence in the following manner:

we first computed the mean pixel value corresponding to each pixel unit with coordinates $(i,j)$ in all the drift corrected images. This is defined as $I_{mean}(i,j) = \frac{1}{N} \sum_{l=1}^{N} I_0(i,j,l)$, where $I_0(i,j,l)$ is the intensity of a pixel $(i,j)$ in each time frame $l$ and $N$ is the number of time frames. While stationary particles will still show up in the mean image $I_{mean}(i,j)$, moving particles on the other hand will have their intensities reduced. Next, we updated each image frame using the following formula

$$I_c(i,j,k) = \max\left[0, I_0(i,j,k) - I_{mean}(i,j)\right] \tag{1}$$

Here, we have defined the corrected intensity $I_c(i,j,k)$ for each time-frame by subtracting the mean intensity $I_{mean}(i,j)$ from the actual intensity value. Note that we set the pixel value to 0 if mean frame intensity value is bigger. As a result of this correction, all stationary particles will disappear from each time frame, while the moving particles will have their intensities reduced as shown below. Note that the total intensity can be written as $I_0(i,j,k) = S(i,j) + M(i,j,k)$, where $S(i,j)$ is the static intensity and $M(i,j,k)$ is the time-dependent fluorescent intensity which captures the movement of clathrin-coated vesicles. If there is no time variation in intensity within a pixel $(i,j)$, then $I_c(i,j,k) = 0$, for that pixel following *Equation 1*. For cases where the time variation is non-zero, we can rewrite *Equation 1* as:

$$I_c(i,j,k) = I_0(i,j,k) - \frac{1}{N} \sum_{l=1}^{N} I_0(i,j,l) = M(i,j,k) - \frac{1}{N} \sum_{l=1}^{N} M(i,j,l). \tag{2}$$

*Equation 2* can be further re-written as,

$$I_c(i,j,k) = \frac{(N-1)}{N} M(i,j,k) - \frac{1}{N} \sum_{\substack{l=1 \\ l \neq k}}^{N} M(i,j,l) = \frac{(N-1)}{N}\left[M(i,j,k) - M_{mean}(i,j,k \neq l)\right] \tag{3}$$

Since the above equation only contains time-dependent intensity, the contribution of it to a pixel $(i,j)$ from all frames $\neq k$ , will mostly be zero and as a result the quantity $M_{mean}\left(i,j,k \neq l\right)$ is expected to be a small quantity. We can thus approximate *Equation 3* as

$$I_c\left(i,j,k\right) \approx \frac{(N-1)}{N}M\left(i,j,k\right) \qquad (4)$$

Due to the correction by the factor $M_{mean}\left(i,j,k \neq l\right)$, there is a decrease in the overall intensity of the moving particles. This decrease of intensity globally will not result in change in magnitudes of flow vectors during the PIV analysis. However, there will be a slight increase in the occurrence of errors while calculating the flow vectors. For this reason, we performed post-processing using standard deviation and median filters. Also note that the intensity correction method will lead to large errors if the intensity dynamics is very slow such that a given pixel is lit up for a large number of time frames, leading to a significant reduction in the estimated value $I_c\left(i,j,k\right)$. To further test the effectiveness of this algorithm, we tested this correction on a movie which does not have any static intensity. We found that while there a slight decrease in the overall intensity, the spatial distribution of the intensity profile in each time frame was invariant.

## Acknowledgements

This work was supported by funding to DS and AM from an NCCS Intramural Collaborative grant NCCS/DIR/2018/24. SP acknowledges funding support from the Department of Science and Technology, Govt. of India (CRG/2022/001891). AM acknowledges funding from Wellcome Trust-DBT India Alliance (IA/I/13/2/501030), and Department of Biotechnology, Govt of India (BT/PR25893/GET/119/174/2017). VA is supported by funding from EMBL Australia. We thank Stefania Marcotti for sharing the PIV code. We thank UB Pandey, DE Rincon Limas, PF Funez and T Littleton for sharing fly lines. We thank Avinash Kshirsagar for help with experiment 4 F Fly stocks from the Bloomington *Drosophila* Stock Centre (NIH P40OD018537) were used in this study. DS thanks AVR. AM thanks the spirit of DFHAN for inspiration.

## Additional information

### Funding

| Funder | Grant reference number | Author |
|---|---|---|
| National Centre for Cell Science | NCCS/DIR/2018/24 | Amitabha Majumdar Deepa Subramanyam |
| Government of India | CRG/2022/001891 | Shivprasad Patil |
| Wellcome Trust/DBT India Alliance | IA/I/13/2/501030 | Amitabha Majumdar |
| Government of India | BT/PR25893/GET/119/174/2017 | Amitabha Majumdar |
| EMBL Australia | | Vaishnavi Ananthanarayanan |

The funders had no role in study design, data collection and interpretation, or the decision to submit the work for publication.

### Author contributions

Surya Bansi Singh, Conceptualization, Resources, Data curation, Formal analysis, Validation, Investigation, Visualization, Methodology, Writing – original draft, Writing – review and editing; Shatruhan Singh Rajput, Conceptualization, Resources, Formal analysis, Validation, Investigation, Visualization, Methodology, Writing – original draft, Writing – review and editing; Aditya Sharma, Resources, Formal analysis, Investigation, Visualization, Methodology, Writing – original draft, Writing – review and editing; Sujal Kataria, Resources, Data curation, Formal analysis, Investigation, Methodology, Writing

– original draft, Writing – review and editing; Priyanka Dutta, Formal analysis, Investigation, Visualization, Methodology, Writing – original draft, Writing – review and editing; Vaishnavi Ananthanarayanan, Data curation, Formal analysis, Investigation, Visualization, Methodology, Writing – original draft, Writing – review and editing; Amitabha Nandi, Conceptualization, Resources, Data curation, Software, Formal analysis, Investigation, Visualization, Writing – original draft, Writing – review and editing; Shivprasad Patil, Conceptualization, Resources, Data curation, Formal analysis, Supervision, Investigation, Visualization, Methodology, Writing – original draft, Writing – review and editing; Amitabha Majumdar, Conceptualization, Resources, Data curation, Formal analysis, Funding acquisition, Investigation, Methodology, Writing – original draft, Writing – review and editing; Deepa Subramanyam, Conceptualization, Resources, Data curation, Formal analysis, Supervision, Funding acquisition, Validation, Investigation, Visualization, Methodology, Writing – original draft, Project administration, Writing – review and editing

### Author ORCIDs
Surya Bansi Singh ⬚ https://orcid.org/0000-0002-6612-5020
Priyanka Dutta ⬚ https://orcid.org/0000-0002-8490-5813
Vaishnavi Ananthanarayanan ⬚ https://orcid.org/0000-0003-2936-7853
Amitabha Nandi ⬚ https://orcid.org/0000-0002-6688-0237
Amitabha Majumdar ⬚ https://orcid.org/0000-0002-6594-0672
Deepa Subramanyam ⬚ https://orcid.org/0000-0002-1650-5690

### Decision letter and Author response
Decision letter https://doi.org/10.7554/eLife.98363.sa1
Author response https://doi.org/10.7554/eLife.98363.sa2

---

## Additional files

### Supplementary files
• MDAR checklist
• Supplementary file 1. Table showing the pixel and frame rate information for each cell type.
• Source data 1. Source data for the figures included in the manuscript.
• Source data 2. Source data for PIV plots included in this manuscript.

### Data availability
All data generated or analysed during this study are included in the manuscript and supporting files.

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
