## [Editor Report]

Singh et al. present an important manuscript showing that the aggregation of mutant Huntingtin in Huntington's disorder affects clathrin-mediated endocytosis, actin organization, and cellular stiffness in non-neuronal cells, which can be partially restored by overexpressing actin-interacting proteins like Hip1 or Arp3. Hip1 or Arp3 also rescue neurodegeneration driven by mutant forms of Huntingtin. The study provides interesting insights into the interplay between Huntingtin aggregation and the biomechanics of endocytosis. The methods and data present compelling support of these findings, although there is a need for further additional analyses in this key area.

---

## [Decision Letter]

[Editors' note: this paper was reviewed by Review Commons.]

---

## [Author Response]

We thank all the reviewers for their comments on our manuscript. We have attempted to address all the points raised by the reviewers and are happy to note that the manuscript is significantly strengthened with the additional experiments that we have performed and from significant restructuring of the manuscript.

Reviewer #1:Major Comments1. The choice of cells looks confusing. *Drosophila* are indeed widely used in research of neurodegeneration mechanisms, since they well reflect the behavioral characteristics of a wide range of brain diseases, but why authors used insect immune cells to study the effect of mHTT on cellular processes? Huntington's disease has a well-established site of origin, in the spiny neurons of the striatum, and they certainly have a different protein context than in insect cells.

We thank the reviewer for this comment. Patients with Huntington’s disorder display a variety of symptoms affecting peripheral, non-neuronal cells, including alterations in the function of immune cells. Hemocytes isolated from *Drosophila* expressing pathogenic forms of Huntingtin also display altered immune responses. Through our manuscript we explore the effect of Huntingtin aggregates on cellular functions of hemocytes. Additionally, we have now included data showing that we are able to observe similar phenotypes in mammalian cells such as neuronal SHSY5Y and HEK293T (Supp. Figure 3). This is indicative of similar effects being exerted by Huntingtin aggregates across cell types and organisms. Finally, we demonstrate that we are able to rescue neurodegeneration in the fly eye upon overexpression of either Hip1 or components of the Arp2/3 complex (Figure 4F), further solidifying our results that Huntingtin aggregates alter CME in an actin-dependent manner and that this largely is responsible for the toxicity. This validates our observations that effects on CME appear to be independent of cell type and that non-neuronal cells such as hemocytes can also be used to study the effects of pathogenic aggregates.

2. The interrelationship between mutant huntingtin and actin cytoskeleton and clathrin-mediated endocytosis that are convincingly demonstrated in other earlier studies in the m/s are described in rather morphological level and there is no description of molecular interactions of proteins belonging to three systems considered, Htt (control vs mutant), actin cytoskeleton and CME. Lack of these data renders the morphological observations unsupported

Previous data shown in Hosp et al., 2017 indicates that a large number of proteins involved in both actin remodelling and clathrin mediated endocytosis are sequestered within Huntingtin aggregates. While the mechanism of sequestration remains unknown, is has also been observed that loss of Huntingtin results in altered organization of the actin cytoskeleton. We have now added points discussing this in the Results section.

3. Three last figures of total eight demonstrate the effect of proteins, responsible for the initiation of certain neurodegenerative pathologies, on the activity of clathrin-mediated endocytosis, and on the properties of actin cytoskeletal system, however neither in the abstract nor in the introduction there is no any word about these proteins; in the discussion only a few words are devoted to one of these proteins TDP-43. When starting the article, did the authors plan to enter this data into the manuscript?

We have now amended this by revising the abstract and the text.

4. It is important to work on the style of the manuscript, the article is difficult to read, it is a collection of data that does not seem related to each other.

We have reorganized the manuscript and have improved on the flow to make it easier for the reader. We apologize for the rather tedious and confusing flow in the previous draft.

Reviewer #2:This manuscript endeavours to explore the link between mutant Huntingtin, clathrin-mediated membrane transport and the actin cytoskeleton: both its dynamics and overall mechanics. As I read it, it carries the interesting idea that pathogenic protein aggregates alter actin cytoskeletal dynamics by sequestering Arp2/3 nucleator. This has two consequences in the authors' experiments: disruption of clathrin-coated vesicle movement and an increase in cellular stiffness. An interesting question is whether these two effects are related: Is the disruption of vesicular movement due to the change in cytoplasmic stiffness? Or could they be features that both reflect the underlying change in actin dynamics. This may be hard to tease apart and beyond the purview of this manuscript.I have some suggestions that could strengthen the MS.Major Comments1. Further characterizing Arp2/3 sequestration. The notion seems to be that actin nucleators would be sequestered (and inactivated) by mutant protein aggregates, as supported by co-localization studies. In addition, could the authors:a) Test if the dynamics of Arp2/3 are altered, comparing e.g. Arp3-GFP FRAP in the aggregates vs that elsewhere.

We indeed attempted the FRAP experiment. However due to some technical difficulties we were not convinced by the extent of FRAP in the transgenic fly line. It appeared as an artifact and we were not comfortable including the data in the manuscript. We have instead provided example files for the reviewer to examine.

b) Test more directly if actin nucleation is altered in cells that have pathogenic mutant aggregates. This could be done by barbed-end labelling (e.g. measuring incorporation of labelled actin in live cells that are lightly permeabilized with saponin).

We have performed barbed-end labeling for HTT Q15 and HTT Q138 expressing cells. Images and quantification have now been added to the revised manuscript as Figures 2H and 2I. While this was a challenging experiment, it was deeply satisfying to observe such dramatic changes indicating a change in the state of the actin cytoskeleton.

2. Does manipulating actin nucleation alter cellular mechanics as it does for clathrin-coated vesicle transport? For example, does inhibition of Arp2/3 (e.g. with CK666) increase cellular stiffness and would stiffness be amelioriated in mutant cells if Arp3 is overexpressed?

We have used LatA to look at whether alteration in the actin cytoskeleton affects cellular stiffness. We found that disruption of the actin cytoskeleton leads to a decrease in cellular stiffness in WT as well as in HTT Q138 expressing cells (shown in Figure 5 and discussed in the Results section). We have also now performed AFM on CK666 treated cells and showed that treatment of CK666 leads to a decrease in cellular stiffness similar to LatA treated cells. This further strengthens our hypothesis that a ‘Goldilocks’ state of actin remodeling and consequently cellular stiffness is required for CME to proceed. We have not performed AFM on cells overexpressing Arp2/3 in HTT Q138 background. However, we believe that it will rescue cellular stiffness as overexpression of Arp2/3 rescues filopodia formation in HTT Q138 expressing cells (Figure 4E) as well as neurodegeneration. AFM data obtained from CK666 treated cells is now added in Supplementary figure 8.

3. Although it may be difficult to determine if the defect in vesicle transport is due to the change in rheology, I wonder if the authors could reinforce their analysis by showing the overall relationship between the two features. It would be interesting if they could plot CCV velocity against elasticity for all the various conditions that they have tested. Would this cumulative analysis be informative?

This data is already present across the manuscript as part of different figures. We are not sure whether we can reuse the same data to put it as part of a different figure which plots the relationship between elasticity and CCS velocity. We would be grateful for advice on whether this is allowed and how to mention that the data is also part of different figures.

4. Focus of the MS. I think that the MS is a little longer and more discursive than it needs to be. I rather struggled to find the focus of the story (which could well be me). There is a deal of repetition that could be profitably cut (the reader may actually find it easier to follow). As well, some anticipation and summaries could be shortened. The final paragraph of the introduction largely summarizes the paper; it could be shortened quite considerably, so that the reader can get directly into the Results themselves. Similarly, the final paragraph of the results is a summary which could work better elsewhere – perhaps, e.g. at the beginning of the discussion.

We have now trimmed and rearranged the text in the manuscript. We have reorganized the manuscript and have improved on the flow to make it easier for the reader. We apologize for the rather tedious and confusing flow in the previous draft. We are open to further suggestions to improve the writing style.

Specific pointsi) Figure 3E. The changes in F-actin flow revealed by PIV are quite dramatic. How reproducible are these changes. (The data presented were from single cells?)

Changes in F- actin flow obtained from PIV analysis (now figure 2J, 7E in MS) were performed on atleast 5 cells of each type, and the results were observed to be consistent across all. The representative figure is a true representative of the data observed.

ii) If TPD43, does it also affect Arp2/3?

We thank the reviewer for this comment. Unfortunately**,** we could not perform this experiment due to the unavailability of a fluorescently tagged TDP43 fly line which which would enable us to visualize whether Arp3 was sequestered within the aggregates.

Minor points:a) Figure 5a, b – why change the order on the x-axis?

We have fixed this now. We have removed figure 5b, since, in the revised MS we are only talking about stiffness instead of viscoelastic properties of the cells.

Overall, I think that the significance of the MS lies in its evidence that sequestration of actin nucleators may be a key effect of mutant protein aggregation, with implications for cellular function. This would provide a useful conceptual framework to understand the cell biological consquences of creating pathogenic protein aggregates.Reviewer #3:SummaryIn the paper, the authors showed that huntingtin aggregates, which play a critical role in initiating neurodegenerative diseases, impair clathrin-mediated endocytosis (CME). Using live cell imaging and AFM, the authors demonstrated that CME is affected by the alteration in actin cytoskeletal organization and cellular viscosity. Further, the authors concluded that there was a strong link between dynamic actin organization and functional CME in the context of neurodegeneration. While the data is interesting and novel, the study in its current form needs major revision before it is accepted.Major comments:1) Figure 2: The authors should show the compromised actin cytoskeleton structure after Lat A and cytoD treatment to back up the findings.

We have included the representative micrographs of compromised cytoskeleton in terms of filopodia formation upon treatment of LatA and CytoD in Supplementary figure 3E.

2) Figure 2g and 2h: Quantification data of filopodia must be supported with representative images.

Figure number has been changed to 2D and 2E. Representative image for the quantification of filopodia has been now included in supplementary figure 3D.

3) RNAi studies must be performed using control siRNA to check off-target effects.

Luc VAL10 was used as a control for all the RNAi experiments. However, data for RNAi is not shown as the phenotype for Luc VAL10 was comparable to WT. We have included Luc VAL10 as a control for Profilin RNAi in the FRAP experiment (Supplementary figure 4C).

4) The result section needs to be reorganized to maintain flow. In the current format, the results of a similar set of experiments are spread across different figures, making it a bit difficult to understand.

We apologize for the inconvenience. This issue has been addressed now.

5) Figure 3d: The expression level and spatial distribution of HTTQ138 transfection were not convincing compared to the httQ15 expression level and the distribution.

Figure 3D (Figure 3A in this MS) shows the data obtained from hemocytes isolated from third instar larva of the same age. These are not transfected cells and are obtained from *Drosophila* larvae using the same Gal4 driver, Cg-Gal4. Thus, the level of expression will be same. However, the distribution may show a change due to the aggregating nature of HTT Q138, while HTT Q15 is non-aggregating and therefore remains diffused.

6) Suppl Figure 2a data must be supported using images showing myosin VI distribution in wild-type vs. HTTQ138 transfected cells.

This data (Supplementary figure 4D in present MS) has been obtained from genetic knockdown of myosin VI. The aim of the experiment was to show that we see similar effects on CCSs movement as we see upon disruption of the actin cytoskeleton.

7) Suppl movie videos are not labeled correctly in the source. It is not possible to locate them and know which videos are referred to in the manuscript.

We apologize for the inconvenience. This issue has been fixed now.

8) Page 8: How do HTT aggregates sequester the actin-binding proteins? An explanation should be provided in the result section.

Previous data shown in Hosp et al., 2017 indicates that a large number of proteins involved in both actin remodelling and clathrin mediated endocytosis are sequestered within Huntingtin aggregates. While the mechanism of sequestration remains unknown, the types of proteins involved in actin remodelling are diverse and do not represent specific types or classes. We have now added points discussing this in the Results section.

9) Page 10: The authors concluded that "increasing the availability of proteins involved in actin reorganization is capable of restoring CME even in the presence of pathogenic aggregates." Since several actin-associated proteins are involved in actin reorganization, which types/classes of proteins are involved in CME restoration? The authors should expand it in the discussion.

As we have only investigated the roles of Hip1 and the Arp2/3 complex, we are confident of only reporting their roles in the context of this manuscript. However, previous data shown in Hosp et al., 2017 indicates that a large number of proteins involved in both actin remodelling and clathrin mediated endocytosis are sequestered within Huntingtin aggregates. While the mechanism of sequestration remains unknown, the types of proteins involved in actin remodelling are diverse and do not represent specific types or classes. Therefore this indicates that modulation of actin, through the sequestration of proteins involved in this process is affected in the presence of Huntingtin aggregates. We have added points detailing this in the results and Discussion sections.

10) The schematic of the proposed model depicting critical steps by which pathogenic proteins inhibit CME is required. It will help readers to understand the molecular mechanism easily.

We have now included a model in the manuscript (Figure 8B).

Minor:1) Figure panel referencing in the text needs to be more consistent, for example, Figure 3e is referred to before Figure 3d., and Figure 2 panels are referred to before Figure 3 panels.

We have reordered the figures and maintained a consistent order throughout.

2) The authors should use similar phrasing throughout the manuscript to avoid confusion. For instance, either use 'HTTQ138' or 'htt Q138'.

We apologize for this. We have now maintained uniform nomenclature through the text.

3) Page 10: AFM indentation experimental part and its discussion in the result section is unnecessary. Shift it to the 'Materials and Method' section.

We have now trimmed this portion and we are now only showing elasticity data and not viscoelasticity.

4) This statement looks a bit exaggerated. There is not sufficient evidence to support the statement- "It can be said that the cells in general behave like a soft glass. The presence of aggregates lowers the effective temperature pushing it nearer to the glass transition, affecting transport."

We have now removed all figures resulting from an analysis that assumes glassy behaviour. Instead, we have now provided a more conventional and well-established analysis to obtain Young’s modulus of cells exhibiting different transport properties.

5) Page 12: What is the basis for selecting proteins Aβ-42, FUSR521C, αSynA30P, αSynA53T, and TDP-43 over other proteins? An explanatory sentence must be added to support the selection.

We have modified the text to clarify this point.